# Diffusion Hyperfeatures: Searching Through Time and Space for Semantic Correspondence

**Grace Luo**[1]   **Lisa Dunlap**[1]   **Dong Huk Park**[1*]   **Aleksander Holynski**[1,2*]   **Trevor Darrell**[1*]

[1]UC Berkeley     [2]Google Research

## Abstract

Diffusion models have been shown to be capable of generating high-quality images, suggesting that they could contain meaningful internal representations. Unfortunately, the feature maps that encode a diffusion model's internal information are spread not only over layers of the network, but also over diffusion timesteps, making it challenging to extract useful descriptors. We propose Diffusion Hyperfeatures, a framework for consolidating multi-scale and multi-timestep feature maps into per-pixel feature descriptors that can be used for downstream tasks. These descriptors can be extracted for both synthetic and real images using the generation and inversion processes. We evaluate the utility of our Diffusion Hyperfeatures on the task of semantic keypoint correspondence: our method achieves superior performance on the SPair-71k real image benchmark. We also demonstrate that our method is flexible and transferable: our feature aggregation network trained on the inversion features of real image pairs can be used on the generation features of synthetic image pairs with unseen objects and compositions. Our code is available at https://diffusion-hyperfeatures.github.io.

## 1   Introduction

Much of the recent progress in computer vision has been facilitated by representations learned by deep models. Features from ConvNets [52, 5, 41, 39], Vision Transformers [9, 2], and GANs [45, 26] have demonstrated great utility in a number of applications, even when compared to hand-crafted feature descriptors [24, 7]. Recently, diffusion models have shown impressive results for image generation, suggesting that they too contain rich internal representations that can be used for downstream tasks. Existing works that use features from a diffusion model typically select a particular subset of layers and timesteps that best model the properties needed for a given task (e.g., features for semantic-level correspondence may be most prevalent in middle layers, whereas textural content may be at later layers). This selection not only requires a laborious discovery process hand-crafted for a specific task, but it also leaves behind potentially valuable information distributed across other features in the diffusion process.

In this work, we propose a framework for consolidating all intermediate feature maps from the diffusion process, which vary both over scale and time, into a single per-pixel descriptor which we dub *Diffusion Hyperfeatures*. In practice, this consolidation happens through a feature aggregation network that takes as input the collection of intermediate feature maps from the diffusion process and produces as output a single descriptor map. This aggregation network is interpretable, as it learns mixing weights to identify the most meaningful features for a given task (e.g., semantic correspondence). Extracting Diffusion Hyperfeatures for a given image is as simple as performing the diffusion process for that image (the generation process for synthetic images, and inversion for real images) and feeding all the intermediate features to our aggregator network.

---

[*]Equal advising contribution.

37th Conference on Neural Information Processing Systems (NeurIPS 2023).

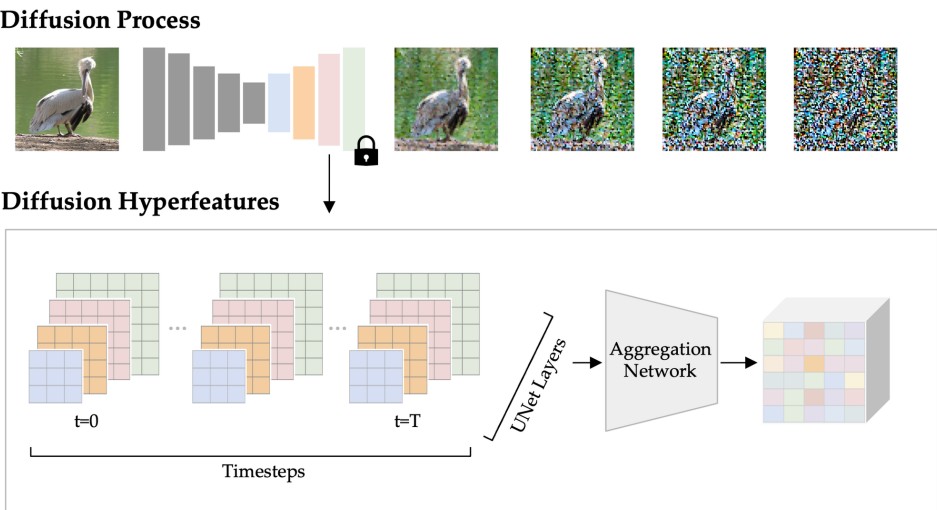

Figure 1: Unlike prior work that hand-selects a subset of raw diffusion features, we extract all feature maps from the diffusion process, varying across both timesteps and layers, and use a lightweight aggregation network to consolidate them into Diffusion Hyperfeatures. For real images, we extract these features from the inversion process, and for synthetic images we extract these features from the generation process. Given a pair of images, we find semantic correspondences by performing a nearest-neighbor search over their Diffusion Hyperfeatures.

We evaluate this approach by training and testing our descriptors on the task of semantic keypoint correspondence, using real images from the SPair-71k benchmark [28]. We present an analysis of the utility of different layers and timesteps of diffusion model features. Finally, we evaluate our trained feature aggregator on synthetic images generated by the diffusion model and show that our Diffusion Hyperfeatures generalize to out-of-domain data.

## 2 Related Work

**Hypercolumn Features.** The term *hypercolumn*, originally from neuroscience literature [18], was first coined for neural network features by Hariharan *et al.* [15] to refer to the set of activations corresponding to a pixel across layers of a convolutional network, an idea that has also been studied in the context of texture [25], optical flow [46], and stereo [20]. One central idea of this line of work is that for precise localization tasks such as keypoint detection and segmentation [15, 50], it is essential to reason at both coarse *and* fine scales rather than the typical coarse-to-fine setting. The usage of hypercolumns has also been popular for the task of semantic correspondence, where approaches must be coarse enough to be robust to illumination and viewpoint changes and fine enough to compute precise matches [27, 30, 44, 29, 1]. Our work revisits the idea of hypercolumns to leverage the features of a recently popular network architecture, diffusion models, which primarily differ from prior work in that the underlying feature extractor is trained on a generative objective and offers feature variation along the axis of time in addition to scale.

**Deep Features for Semantic Correspondence.** There has been a recent interest in transferring representations learned by large-scale models for the task of semantic correspondence. Peebles *et al.* [34] addressed the task of congealing [23] by supervising a warping network with synthetic GAN data produced from a learned style vector representing shared image structure. While prior work [34, 43] has demonstrated that generative models can produce aligned image outputs, which is useful supervision for semantic correspondence, we study how well the raw intermediate representations themselves would perform for the same task. Utilizing self-supervised representations, in particular from DINO [4], has also been especially popular. Prior work has demonstrated that it is possible to extract high-quality descriptors from DINO that are robust and versatile across domains and challenging instances of semantic correspondence [2]. These descriptors have been used for downstream tasks such as semantic segmentation [14], relative camera pose estimation [10], and dense visual alignment [31, 11]. While DINO does indeed contain a rich visual representation,

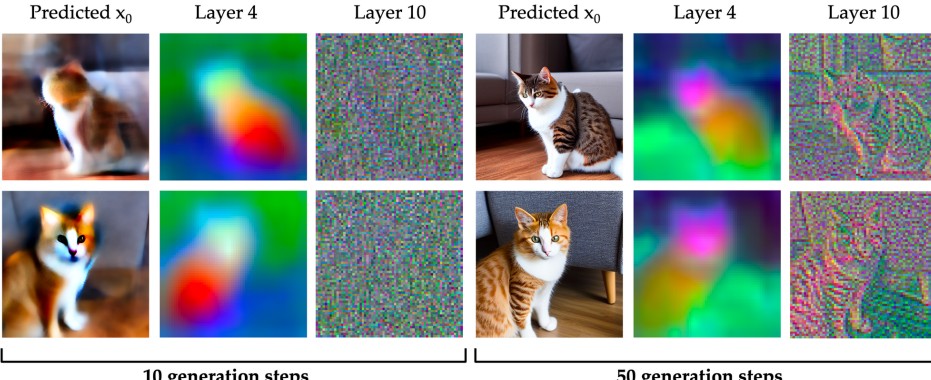

*Cat sitting in a living room.*

| Predicted $x_0$ | Layer 4 | Layer 10 | Predicted $x_0$ | Layer 4 | Layer 10 |

**10 generation steps**      **50 generation steps**

Figure 2: We show an example pair of synthetic images for the prompt "cat sitting in a living room" and the PCA of the features from Layers 4, 10 during both an early and late generation step. While different layers capture different image characteristics (here Layer 4 delineates the face vs. body and Layer 10 captures the edges), these features also evolve and become more fine-grained over time.

diffusion features are under-explored for the task of semantic correspondence and likely contain enhanced semantic representations due to training on image-text pairs.

**Diffusion Model Representations.** There have been a few works that have analyzed the underlying representations in diffusion models and proposed using them for downstream tasks. Plug-and-Play [43] injects intermediate features from a single layer of the diffusion UNet during a second generation process to preserve image structure in text-guided editing. FeatureNeRF [49] distills diffusion features from this same layer into a neural radiance field. DDPMSeg [3] and ODISE [48] aggregate features from a hand-selected subset of layers and timesteps for semantic and panoptic segmentation respectively. While these works also consider a subset of features across layers and/or time, our work primarily differs in the following ways: first, rather than hand-selecting a subset of features, we propose a learned feature aggregator building on top of Xu *et al.* [48] that weights all features and distills them into a concise descriptor map of a fixed channel size and resolution. Furthermore, all of these methods solely use the generation process to extract representations, even for real images. In contrast, we use the inversion process, where we are able to extract higher-quality features at these same timesteps, as seen in Figure 3.

## 3 Diffusion Hyperfeatures

In a diffusion process, one makes multiple calls to a UNet to progressively denoise the image (in the case of generation) or noise the image (in the case of inversion). In either case, the UNet produces a number of intermediate feature maps, which can be cached across all steps of the diffusion process to amass a set of feature maps that vary over timestep and layers of the network. This feature set is rather large and unwieldy, varying in resolution and detail across different axes, but contains rich information about texture and semantics. We propose using a lightweight aggregation network to learn the relative importance of each feature map for a given task (in this paper we choose semantic correspondence) and consolidate them into a single descriptor map (our Diffusion Hyperfeatures). To assess the quality of this descriptor map, we compute semantic correspondences for an image pair, by using a nearest-neighbors search on the extracted descriptors, and compare these correspondences to the ground-truth annotations [28, 47]. An overview of our method is shown in Figure 1.

Our approach is composed of two core components. Extraction (Section 3.1): We formulate a simplified and unified extraction process that accounts for both synthetic and real images, which means we are able to use the same aggregation network on features from both image types. Aggregation (Section 3.2): We propose an interpretable aggregation network that learns mixing weights across the features, which highlights the layers and timesteps that provide the most useful features unique to the underlying model and task.

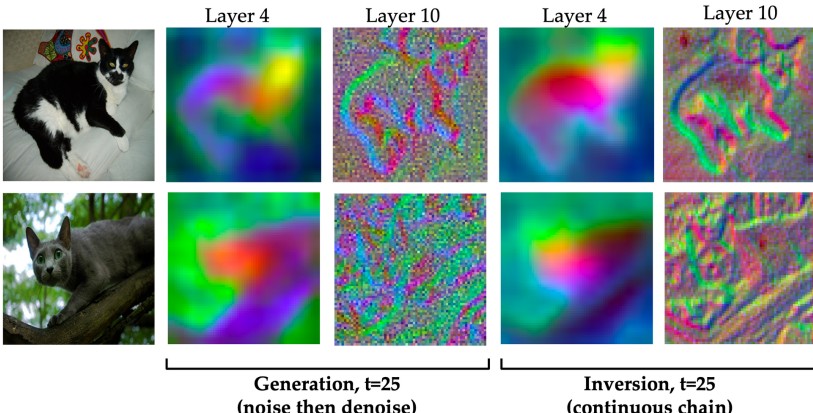

|   | Layer 4 | Layer 10 | Layer 4 | Layer 10 |
|---|---|---|---|---|

**Generation, t=25**
**(noise then denoise)**

**Inversion, t=25**
**(continuous chain)**

Figure 3: We show an example pair of real images from SPair-71k and the PCA of the features from Layers 4, 10 when extracted at the middle timestep $t = 25$. While prior work extracts generation features by noising and denoising the image independently at the specific timestep (left), in our approach we extract inversion features from one continuous chain (right). Extracting features from the same timestep of the inversion chain can produce features more true to original image content.

## 3.1 Diffusion Process Extraction

One popular sampling procedure for a trained diffusion model is DDIM sampling [40] of the form

$$x_t = \sqrt{\alpha_t}x_0 + \sqrt{1-\alpha_t}\epsilon_t \text{ where } \epsilon_t \sim \mathcal{N}(0,1)$$

where $x_0$ is the clean image, $\epsilon_t$ is the noise prediction from the diffusion model conditioned on the timestep $t$ and noisy image $x_{t+1}$ from the previous timestep, and $x_t$ is the prediction for the next timestep. To run generation, one runs the reverse process from $t = T$ to 0, with the input $x_T$ set to pure noise sampled from $\mathcal{N}(0,1)$. To run inversion, one runs the forward process from $t = 0$ to $T$, with the input $x_0$ set to the clean image.

**Generation.** When synthesizing an image, we can cache the intermediate feature maps across the generation process, which already contain shared representations that can be used to relate the image to other synthetic images, as seen in the PCA visualization of the feature maps in Figure 2. In this example, we see that the head and body of the two cats share a corresponding latent representation throughout almost the entire generation process, even as early as the first ten steps, where the inputs to the UNet are almost pure noise. This can be explained by the preliminary prediction for the final image $x_0$, which already lays out the general idea of the image including the structure, color, and main subject. As generation progresses, the principal components of the features also evolve, with Layer 4 changing from displaying coarse to more refined common semantic sub-parts and Layer 10 changing from displaying no shared characteristics to high-frequency image gradients. These observations indicate that the diffusion model provides coarse and fine features that capture different image characteristics (i.e. semantic or textural information) throughout different combinations of layers and timesteps. Hence, we find it important to extract features from all layers and timesteps in order to adequately tune our final descriptor map to represent the appropriate level of granularity needed for a given task.

**Inversion.** These same useful features can be extracted for real images through the inversion process. Although inversion is a process that destructs the real image into noise, we observe that its features contain useful information akin to the generation process for synthetic images. In Figure 3, we can see that our inversion features are able to reliably capture the full body of both cats and their common semantic subparts (head, torso, legs) in Layer 4 and their edges in Layer 10 even at a timestep when the input to the model is relatively noisy. In contrast, using the generation process to analyze real images (as done in prior work) leads to hyperparameter tuning and tradeoffs. For example, at timesteps close to $t = T$ where in-distribution inputs are close to noise, the features start to diverge from information present in the real image and may even hallucinate extraneous details, as seen in Figure 3. In this example, because the color of the top cat's stomach is white like the background, the generation features from Layer 4 merge the stomach with the background. Similarly, because there is low contrast between the bottom cat and the background, the generation features from

Layer 10 fail to capture the silhouette of the cat and instead depict random texture edges. Intuitively, inversion features are more trustworthy because of the notion of *chaining*, where at every step the input is some previous output of the model rather than a random mixture of an image and noise, and every extracted feature map is therefore interrelated. Extracting features from a continuous inversion process also induces symmetry with the generation process, which in Section 4.3 we demonstrate allows us to use both feature types interchangeably with the same aggregation network.

## 3.2  Diffusion Hyperfeatures Aggregation

Given a dense set of feature maps from the diffusion process, we now must efficiently aggregate them into a single descriptor map without omitting the information from any layers or timesteps. A naive solution would be to simply concatenate all feature maps into one very deep feature map, but this proves to be too high dimensional for most applications. We address this issue with our aggregation network, which standardizes the feature maps with tuned bottleneck layers and sums them according to learned mixing weights. Specifically, for a given feature map $r$ we upsample it to a standard resolution, pass through a bottleneck layer $B$ [16, 48] to a standard channel count, and weight it with a mixing weight $w$. The final Diffusion Hyperfeatures take on the form

$$\sum_{s=0}^{S} \sum_{l=1}^{L} w_{l,s} \cdot B_l(r_{l,s})$$

where $L$ is the number of layers and $S$ is the number of timesteps. Note that we run the diffusion process for a total number of $T$ timesteps but only select a subsample of $S$ timesteps for aggregation to conserve memory. We opt to share bottleneck layers across timesteps, meaning we use a total of $L$ bottleneck layers. However, we learn $L \cdot S$ unique mixing weights for every combination of layer and timestep. We then tune these bottleneck layers and mixing weights using task-specific supervision. For semantic correspondence, we flatten the descriptor maps for a pair of images and compute the cosine similarity between every possible pair of points. We then supervise with the labeled corresponding keypoints using a symmetric cross entropy loss in the same fashion as CLIP [36]. During training, we downscale the labels according to the resolution of our descriptor maps. When running inference, we upsample the descriptor maps before performing nearest neighbors matching to predict semantic keypoints. We demonstrate that this lightweight parameterization of our aggregation network is performant (see Section 4.1) and interpretable (see Section 4.2) without degrading the open-domain knowledge represented in the diffusion features (see Section 4.3).

## 4  Experiments

**Baselines.** We compare against both zero-shot and supervised methods for the task of semantic correspondence. For our zero-shot baselines, we compare against the DINO [2, 4] Layer 9 key features and DINOv2 [32] Layer 11 token features, derived from a self-supervised model trained on ImageNet [8] or LVD-142M [32] respectively. For our supervised baselines, we compare against DHPF [29] and CATS++ [6], which aggregate ResNet-101 [16] hypercolumn features. The features are derived from a supervised model pretrained on ImageNet and finetuned on SPair-71k [28]. We also compare against two diffusion baselines, selecting a single feature map from a known semantic layer [43] (SD-Layer-4) and concatenating feature maps from all layers (SD-Concat-All).

**Experimental Details.** Our method extracts descriptors from Stable Diffusion v1-5 [37], a generative latent diffusion model trained on LAION-5B [38]. We tune our bottleneck layers and mixing weights on SPair-71k for up to 5000 steps with a batch size of 2 and a learning rate of 1e-3 using an AdamW optimizer [21]. We extract features from the UNet decoder layers (denoted as Layers 1 to 12), specifically the outputs of the residual block before the self- and cross- attention blocks. Our full method runs the inversion or generation process over the standard $T = 50$ scheduler timesteps and subsamples every 5 steps, thereby aggregating features over $S = 11$ total steps. We also ablate using a "one-step" process ($T = 1, S = 1$). When operating on real images, we feed the empty prompt "". We find that extracting features only from the unconditional model removes the need for manual prompting and yields high-quality results. When operating on synthetic images, we extract features only from the conditional model, or the branch conditioned on the prompt that generated the image.

**Metrics.** We report results in terms of PCK@$\alpha$, or the percentage of correct keypoints at the threshold $\alpha$. The predicted keypoint is considered to be correct if it lies within a radius of $\alpha * max(h, w)$ of

| | # Layers $L$ | # Timesteps $S$ | SPair-71k | | CUB | |
|---|---|---|---|---|---|---|
| | | | ↑ PCK@$0.1_{img}$ | ↑ PCK@$0.1_{bbox}$ | ↑ PCK@$0.1_{img}$ | ↑ PCK@$0.1_{bbox}$ |
| DINO [2] | 1 | - | 51.68 | 41.04 | 72.72 | 55.90 |
| DINOv2 [32] | 1 | - | 68.33 | 56.98 | 89.96* | 76.83* |
| DHPF [29] | 34 | - | 55.28 | 42.63 | 77.30 | 61.42 |
| CATS++ [6] | 30 | - | 70.26 | 57.06 | 75.92 | 59.49 |
| SD-Layer-4 | 1 | 1 | 58.80 | 46.58 | 78.43 | 61.22 |
| SD-Concat-All | 12 | 1 | 52.12 | 41.83 | 70.22 | 54.05 |
| **Ours** | 12 | 11 | **72.56** | **64.61** | **82.29** | **69.42** |
| Ours-One-Step | 12 | 1 | 63.74 | 54.69 | 76.59 | 62.11 |
| SD-Layer-Pruned | 1 | 1 | 57.69 | 48.16 | 80.67 | 67.21 |
| Ours-Pruned | 1 | 1 | 64.02 | 53.74 | 79.10 | 63.95 |
| Ours-SDv2-1 | 12 | 11 | 70.74 | 64.85 | 80.39 | 68.04 |

Table 1: We evaluate our semantic keypoint matching performance on real images from SPair-71k and CUB. For our CUB evaluation, we transfer the model tuned on SPair-71k. We compare against Stable Diffusion baselines that extract features from a single layer (SD-Layer-4) or concatenation of all layers (SD-Concat-All). We ablate pruning to the single feature map with the highest mixing weight selected by our method, either as the raw feature map (SD-Layer-Pruned) or after the bottleneck layer (Ours-Pruned). We ablate tuning our method with only one timestep (One-Step) or features from another Stable Diffusion variant (SDv2-1). *Note that DINOv2 was trained on samples from CUB [32].*

the ground truth annotation, where $h, w$ denotes the dimensions of the entire image ($\alpha_{img}$) or object bounding box ($\alpha_{bbox}$) in the original image resolution. Following Amir *et al.* [2] we use the threshold $\alpha = 0.1$ [2]. We use the default input resolution for each baseline, and we "compute the metrics on the standard setting, *i.e.* the original image size" [42] to ensure a fair comparison. For DINO, DHPF, and CATS++ we use the resolutions 224, 240, and 512 respectively. For DINOv2 we use resolution 770 to account for its larger patch size and for Stable Diffusion variants we use resolution 224 in line with our DINO baseline.

**Additional Experiments.** In the Supplemental we provide additional experiments ablating performance of raw features across individual decoder layers and model variants, performance of raw inversion vs. generation features, the effect of aggregating DINO and DINOv2 features, and applying Diffusion Hyperfeatures for dense warping.

## 4.1 Semantic Keypoint Matching on Real Images

We evaluate on the semantic keypoint correspondence benchmark SPair-71k [28], which is composed of image pairs from 18 object categories spanning animals, vehicles, and household objects. Following prior work [2], we evaluate on 360 random image pairs from the test split, with 20 pairs per category. To test the method's ability to transfer to unseen data, we also evaluate on CUB [47], which is composed of semantic keypoints on a variety of bird species. Our validation set is 360 random image pairs taken from the Kulkarni *et al.* [22] validation split.

As seen in Table 1, even when compared with the competitive baselines DINOv2 and CATS++, our method achieves the best result with 72.56 PCK@$0.1_{img}$ and 82.29 PCK@$0.1_{img}$ on SPair-71k and CUB respectively. Selecting a single diffusion feature map (SD-Layer-4) already outperforms DINO and DHPF by a margin of at least 4% in PCK@$0.1_{img}$ , likely due to the underlying model's larger and more diverse training set. Conversely, naively concatenating all feature maps (SD-Concat-All), degrades this improvement, likely because each map captures a different level of granularity and therefore should not be equally weighted for a coarse-level semantic similarity task. When aggregating all feature maps across time and space using our Diffusion Hyperfeatures, we see a significant boost of 14% in PCK@$0.1_{img}$ over SD-Layer-4. This sizeable performance improvement indicates that (1) while a single layer may capture a good deal of semantic information, the other layers also capture complementary information that can be useful for disambiguating points on a finer level and (2) the diffusion model's internal knowledge about the image is not fully captured by a single pass through the UNet but is rather spread out over multiple timesteps.

In Figure 4, we show qualitative examples of our predicted semantic correspondences compared with the strongest baselines. In general, while both baselines are able to relate broad semantic regions

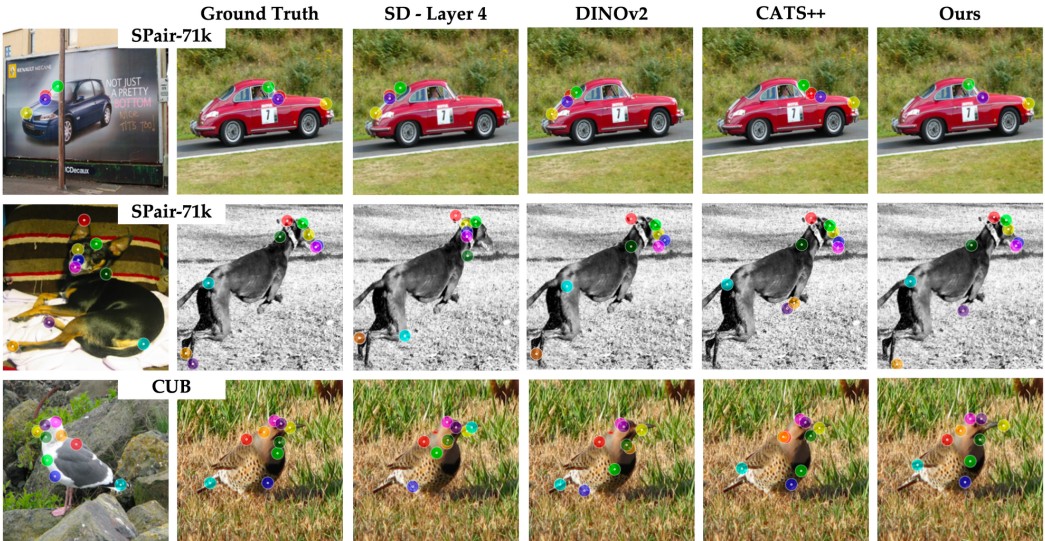

Figure 4: Example images from SPair-71k and CUB, the ground-truth user-annotated correspondences, and predicted correspondences from each method.

such as the head, arms, or legs, they struggle with fine-grained subparts and confuse them with other visually similar regions. For example, SD-Layer-4 and DINOv2 confuse the headlight (yellow) of the blue car with the rear light of the red car, whereas our method is able to correctly reason about the front vs. back of the cars. In a similar fashion, unlike our method both of these baselines also confuse the back of the seagull's neck (orange) with the front of the northern flicker bird's neck.

## 4.2 Ablations

**Number of Diffusion Steps.** We ablate the importance of the time dimension by tuning a variant of our method on feature maps from a one-step inversion process (Ours-One-Step), where there is only one possible timestep to extract from. While aggregating over all layers performs better than single layer selection (SD-Layer-4) or simple concatenation (SD-Concat-All) by a margin of at least 5% in PCK@$0.1_{img}$, it also heavily lags behind our full method. As indicated by our mixing weights in Figure 5, the most useful information for semantic correspondence are concentrated in the early timesteps of the diffusion process, where the input image is relatively clean but contains some noise. Timestep selection can be thought of as a knob for the amount of high frequency detail present in the image to analyze, where at these early timesteps the model is implicitly mapping the noisy input to a smoother image, similar to the oversmoothed cats for the predicted $x_0$ in Figure 2. Evidently not all texture and detail is necessary for the task of semantic correspondence, and the model is able to produce more useful features at different timesteps which highlight different image frequencies.

**Pruning.** Since our method employs interpretable mixing weights, we have a ranking of the relative importance of each layer and timestep combination for the task of semantic correspondence. To verify this ranking, we ablate pruning to the single feature map with the highest mixing weight. As seen in Figure 5, for Ours-SDv1-5 this is the feature map associated with Layer 5, Timestep 10. Referring to Table 1, this automatically selected raw feature map (SD-Layer-Pruned) performs comparably to the feature map manually selected by prior work (SD-Layer-4). Interestingly, if only viewing features from a one-step inversion process it would seem that Layer 5 features are significantly worse than Layer 4 features, as discussed further in the Supplemental, but the story changes after tuning selection over both time and space. After passing this pruned feature map to our learned bottleneck layer, the feature map performs comparably on CUB and 6% better in PCK@$0.1_{img}$ on SPair-71k, as seen when comparing SD-Layer-Pruned and Ours-Pruned in Table 1. This trend validates that our bottleneck layer does not degrade the power of the original representation, but rather refines it in a way that is likely helpful for complex object categories present in SPair-71k beyond birds. Considering the 9% gap in PCK@$0.1_{img}$ between our pruned and full method (Ours-Pruned vs. Ours), it becomes evident that it is not a single feature map that drives our strong performance but rather the soft mixing of many feature maps.

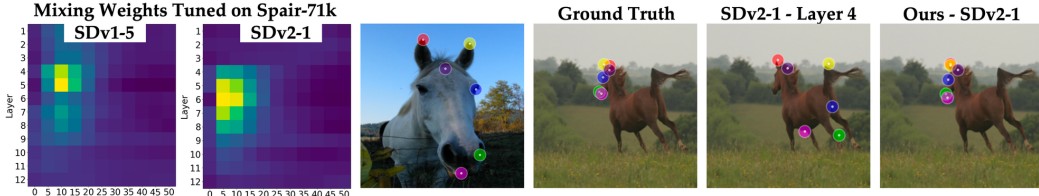

Figure 5: The learned mixing weights when aggregating SDv1-5 vs. SDv2-1 features across multiple layers and timesteps. Bright yellow denotes a high weighting, and dark blue denotes a low weighting. We also depict predicted correspondences from SDv2-1-Layer-4 vs. Ours-SDv2-1. While Layer 4 features from SDv1-5 perform well in semantic correspondence, this same layer in SDv2-1 performs extremely poorly. Our method automatically learns the best layers depending on the model variant.

**Model Variant.** We also ablate using a different model variant for our diffusion feature extraction, namely SDv2-1. Most notably, SDv2-1 differs from SDv1-5 in its large-scale text encoder, which scales from CLIP [36] to OpenCLIP [19]. The mixing weights learned for both model variants, depicted in Figure 5, showcase the same high-level trends, where the features found to be the most useful for semantic correspondence are concentrated in middle Layers 4-9 and early Timesteps 5-15. However, on a more nuanced level, the behavior starts to diverge with regards to the relative importance of layers and timesteps within this range. Namely, layer selection moves from Layers 4-5 to higher resolution Layers 5-7 from SDv1-5 to v2-1. This behavior is confirmed by our early hyperparameter sweeps of raw feature maps across model variants discussed in the Supplemental, where in fact the Layer 4 feature map of SDv2-1 performs extremely poorly for the task of semantic correspondence. Timestep selection also moves from Timestep 10 to Timestep 5 from SDv1-5 to v2-1, which is surprising because this means that SDv2-1 tends to select higher resolution feature maps from timesteps with higher frequency inputs for a task where it is essential to abstract fine-grained details into semantically meaningful matches. These trends seem to imply that a more powerful text encoder produces shared semantic representations at increased levels of detail, possibly because the model is better able to connect more distantly related visual concepts via text instead of giving them completely disjoint representations. Hence, our aggregation network is able to dynamically adjust to the representations being aggregated and the task at hand, both of which influence the most important set of features to select from the diffusion process.

## 4.3 Transfer on Synthetic Images

In addition to evaluating the transfer of our aggregation network to other datasets such as CUB, we also evaluate on synthetic images. Specifically, we take *the same aggregation network tuned on inversion features* and simply flip the timestep ordering to operate on generation features. Therefore in this setting, we are testing our network's ability to generalize (1) to a completely unseen feature type from a different diffusion process (inversion vs. generation) and (2) out-of-domain object categories that are not present in SPair-71k. Surprisingly, our network generalizes well, outperforming predictions from both DINO and the raw feature map from the last step of the generation process (SD-Layer-4) as seen in Figure 6. Although DINO and SD-Layer-4 are generally able to correspond broad semantic regions correctly, they sometimes have difficulty with relative placement of subparts. In the case of the rungs of the Eiffel Tower (purple, pink, green), DINO collapses all of its predictions onto the middle rung and SD-Layer-4 collapses them onto the left rung, whereas our method is able to correctly correspond the middle and side rungs of the tower in a triangle formation. The baselines can also be distracted by other objects in the scene that are visually similar. In the case of the mermaid's legs (red, yellow), both baselines incorrectly correspond certain points with the rock in the right image, whose contours and silhouette resemble the legs. On the other hand, our method is able to predict more reliable correspondences for fine-grained subparts, even in these challenging cases with unseen textures (lego, snow) and categories (hat, cactus). The ability of our aggregation network to extend to open-domain synthetic images opens up the exciting possibility of generating custom synthetic datasets with pseudo ground-truth semantic correspondences, which we demonstrate are more precise than correspondences derived from the raw feature maps.

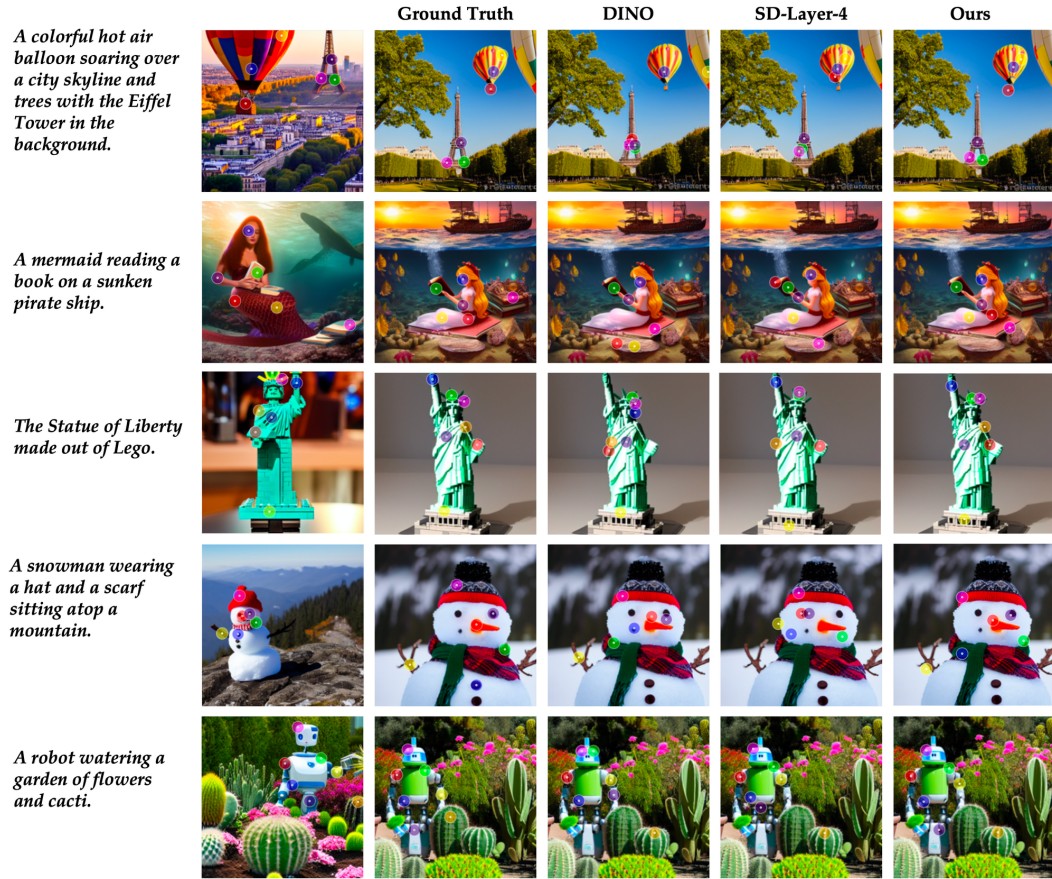

Figure 6: Example synthetic images and their text prompts, the ground-truth user-annotated correspondences, and predicted correspondences from DINO, SD-Layer-4, and our method. We transfer the aggregation network tuned on inversion features of real images to generation features of synthetic images that are completely out-of-domain compared to the SPair-71k categories.

## 5  Conclusion

We demonstrate that our Diffusion Hyperfeatures are able to distill the information distributed across time and space from a diffusion process into a single descriptor map. Our intepretable aggregation network also enables automatic analysis of the most useful layers and timesteps based on both the underlying model and task of interest. We outperform methods that use supervised hypercolumns or self-supervised descriptors by a large margin on a semantic keypoint correspondence benchmark comprised of real images. Although we tune on a small set of real images with limited categories, we demonstrate that our method is able to retain the open-domain capabilities of the underlying diffusion features by demonstrating strong performance in predicting semantic correspondences in challenging synthetic images, especially compared to using the raw feature maps. Our ability to predict high-quality correspondences derived from the same feature maps used to produce the synthetic image could potentially be employed to create synthetic image sets with pseudo-labeled semantic keypoints, which would be valuable for downstream tasks such as image-to-image translation or 3D reconstruction.

## 6  Acknowledgements

We thank Angjoo Kanazawa, Yossi Gandelsman, Norman Mu, David Chan, and Jitendra Malik for helpful discussions. This work was supported in part by DoD including DARPA's SemaFor, PTG and/or LwLL programs, as well as BAIR's industrial alliance programs.

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
