## Supplementary Material

### 6.1    Computational Resources

Our aggregation network takes one day to train on one Nvidia Titan RTX GPU. Inference with our method can run on a Nvidia T4 GPU on a Google Colab notebook.

In Table 2 we compare the memory consumption of our descriptors as well as the inference time against the baselines that also use features from large pretrained models. While our full method explores the upper bound by utilizing features across the diffusion process, which takes 6.62s, one can also use the same pretrained weights to evaluate faster pruned versions of the same model. For these pruned versions, we stop the diffusion process after 1, 5, and 10 timesteps. The version that utilizes the first 10 timesteps performs close to our full method, with a 4% improvement in PCK@$0.1_{img}$ over DINOv2 with an almost 2x faster inference process. Note that the inference times for DINO and DINOv2 are bottlenecked by the log binning algorithm from Amir *et al.* [2] to contextualize the features into descriptors.

| | ↑ PCK@$0.1_{img}$ | Memory per Descriptor | Inference Time per Pair |
|---|---|---|---|
| DINO [2] | 51.68 | 75 MB | 3.02 s |
| DINOv2 [32] | 68.33 | 75 MB | 2.99 s |
| SD-Layer-4 | 58.80 | 10 MB | 0.33 s |
| SD-Concat-All | 52.12 | 1.8 GB | 0.87 s |
| Ours - One-Step | 63.74 | 6MB | 0.27s |
| Ours (1 Timestep) | 64.61 | 6 MB | 0.28 s |
| Ours (5 Timesteps) | 69.28 | 6 MB | 0.86 s |
| Ours (10 Timesteps) | 72.00 | 6 MB | 1.60 s |
| Ours (50 Timesteps) | 72.56 | 6 MB | 6.62 s |

Table 2: We compare average memory and runtime consumption on real images from SPair-71k.

### 6.2    Stable Diffusion Model Variant

In Figure 7, we ablate the behavior of individual raw feature maps from each layer across multiple variants of Stable Diffusion. We extract these features from a one-step inversion process. We report the semantic keypoint matching accuracy on real images from SPair-71k according to PCK@$0.1_{img}$ . Due to limited computational resources, in this experiment we performed nearest neighbor matching on 64x64 resolution feature maps (the maximum possible resolution of Stable Diffusion) and rescaled our predictions to coordinates in the original image resolution. Therefore, we also include a DINO baseline [2] that uses the same procedure as reference for this experimental setting.

Viewing Figure 7, for Stable Diffusion models that share the same broader model variant (e.g., SDv1-3 vs. SDv1-4 vs. SDv1-5), the behavior across layers is similar. In contrast, there is a larger difference in layer behavior when comparing SDv1 (pink) and SDv2 (blue). For SDv1 Layer 4 outshines all other layers, consistent with observations from prior work [43], but this layer actually performs extremely poorly in SDv2. In fact, for SDv2 it is Layers 5 and 6 that are the layers that are strong at semantic correspondence. As seen in Figure 8, SDv2-1's Layer 4 features seem to perform poorly at semantic correspondence because they also strongly encode positionality; while they are able to disambiguate the birds and the backgrounds, they also separate the top left (green), top right (blue), bottom left (orange), and bottom right (pink) of the image. Perhaps SDv2 also encodes positionality in the Layer 4 features because this information is relevant when synthesizing images from prompts that describe relations or more complex object compositions, which SDv1's CLIP struggles with representing [17]. Finally, the behavior when concatenating feature maps from all layers (Concat All) is also very different between SDv1 and SDv2 when viewing Figure 7. While for SDv1 Concat All performs reasonably well, slightly lagging behind its single best feature map, for SDv2 it exhibits subpar performance. This trend is better understood when examining the PCA of these feature maps for two images in Figure 8, where for SDv1-5 Concat All produces a meaningful feature map that delineates the bird, branch, and background and for SDv2-1 Concat All produces a muddy feature map that only delineates the top vs. bottom of the image. This phenomenon where SDv2 produces a low-quality aggregated feature map in the case of simple concatenation is likely

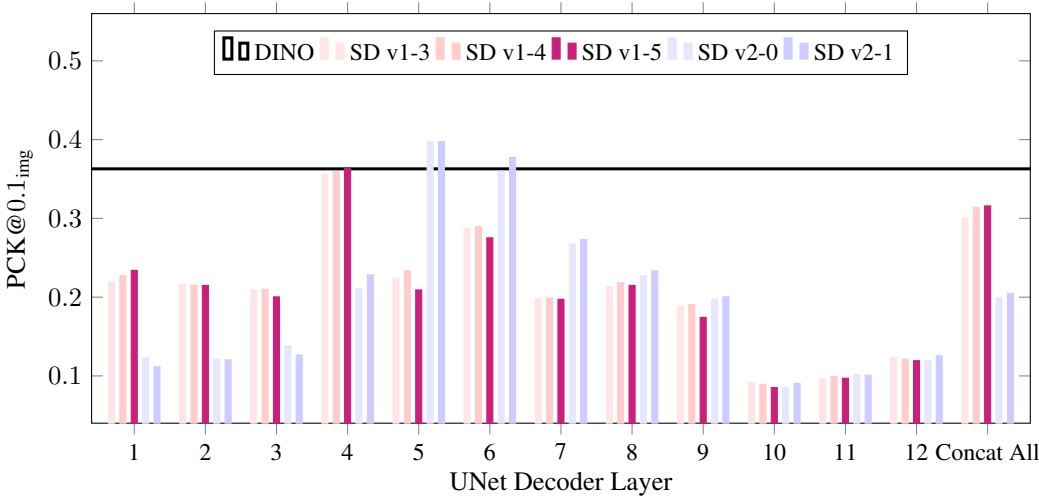

Figure 7: We report the behavior of individual layers across different variants of Stable Diffusion. We extract the raw feature map from a one-step inversion process and compute the semantic keypoint matching performance on real images from SPair-71k.

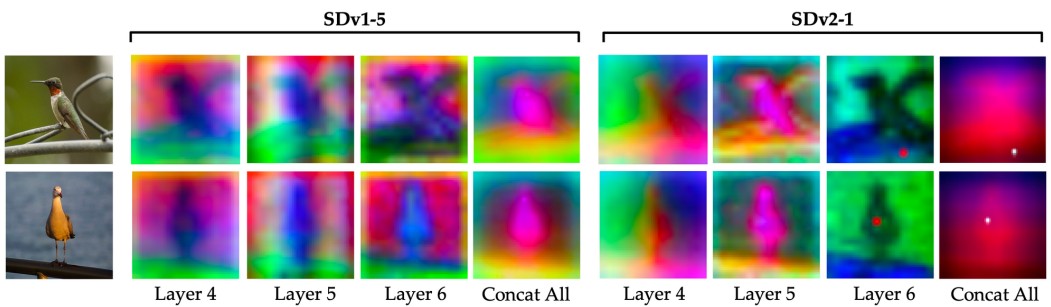

Figure 8: We show an example pair of real images from SPair-71k and the PCA of the features from Layers 4-6 and Concat All extracted from a one-step inversion process for SDv1-5 and SDv2-1.

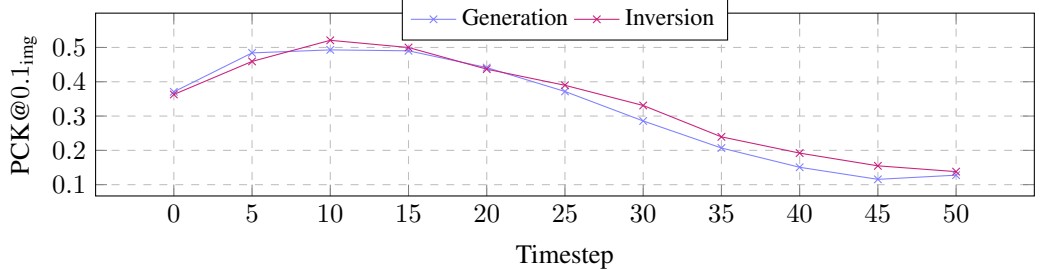

Figure 9: We report the behavior of inversion vs. generation features across timesteps (t=0 denotes a clean image and t=50 denotes pure noise), fixed to the raw feature maps from Layer 4 in SDv1-5. We extract generation features by independently noising and denoising the image at each timestep, and we extract inversion features from one continuous chain. We compute the semantic keypoint matching performance on real images from SPair-71k using the same procedure as Section 6.2.

also because its stronger encoding of positionality dominates the encoding of semantics across the features. On the other hand, our method is able to meaningfully aggregate features across layers for *both* SDv1 and SDv2, as demonstrated by the strong keypoint matching performance from both variants in Table 1. Our method is also able to reflect the differing layer behaviors across different Stable Diffusion variants, as seen by the consistency between the the trends observed in Figure 7 and the learned mixing weights in Figure 5.

## 6.3 Inversion vs. Generation Features

As discussed in Figure 3, we opt to use features from the inversion process rather than the generation process when analyzing real images, because we find that it produces higher quality features at later timesteps when the input to the model looks closer to noise. As seen in Figure 9, the raw feature maps from inversion at these late timesteps ($t = 25$ to $50$) are generally more informative than those from generation for semantic keypoint matching, with a margin of as much as 5% $PCK@0.1_{img}$, while maintaining comparable performance at earlier timesteps ($t = 0$ to $25$).

## 6.4 Additional Evaluation Datasets

We conduct our main evaluation on SPair-71k and CUB because they presented more complex and varied examples than other benchmarks, which are largely composed of simple image pairs with "similar viewpoints and scales" [28]. Nevertheless, in Table 3 we compare our method against the strongest baselines on PF-PASCAL [12] and PF-WILLOW [13], where our method outperforms DINOv2 by 2% and 3% $PCK@0.1_{img}$ respectively.

| | PF-PASCAL | | PF-WILLOW | |
| --- | --- | --- | --- | --- |
| | $\uparrow PCK@0.1_{img}$ | $\uparrow PCK@0.1_{bbox}$ | $\uparrow PCK@0.1_{img}$ | $\uparrow PCK@0.1_{bbox}$ |
| DINOv2 [32] | 84.30 | 78.99 | 86.64 | 71.34 |
| CATS++ [6] | 68.02 | 62.96 | 78.87 | 66.09 |
| **Ours** | **86.67** | **82.85** | **89.61** | **77.98** |

Table 3: We compare our semantic keypoint matching against the strongest baselines on real images from PF-PASCAL and PF-WILLOW.

## 6.5 DINO Features Aggregation

In Table 4, we ablate the effect of single layer selection, naive concatenation, and training an aggregation network for DINO and DINOv2, symmetric to the ablations we performed for our method. Note that in previous experiments for DINOv2 we used inputs of resolution 770 to account for its large patch size, but in this experiment we use the same resolution of 224 across all feature backbones to ensure comparability. Ultimately, our method that trains an aggregation network on top of Stable Diffusion features performs the best at 72.56% $PCK@0.1_{img}$, compared with 54.69% and 68.37% $PCK@0.1_{img}$ for DINO and DINOv2 respectively. Consistent with the hand-selected features explored in Amir *et al.* [2], our aggregation network on top of DINO features learns that Layers 9 - 11 are most useful for the semantic correspondence task.

| | SPair-71k | |
| --- | --- | --- |
| | $\uparrow PCK@0.1_{img}$ | $\uparrow PCK@0.1_{bbox}$ |
| DINO [2] | 51.68 | 41.04 |
| DINO - Concat All | 20.17 | 13.60 |
| DINO + Aggregation Network | 54.69 | 44.29 |
| DINOv2 [32] | 60.14 | 46.94 |
| DINOv2 - Concat All | 60.89 | 47.69 |
| DINOv2 + Aggregation Network | 68.37 | 56.35 |
| SD-Layer-4 | 58.80 | 46.58 |
| SD-Concat-All | 52.12 | 41.83 |
| **Ours** | **72.56** | **64.61** |

Table 4: We ablate the semantic keypoint matching performance of an aggregation network trained on top of DINO features on real images from SPair-71k. To ensure consistency across all backbones, we use the same input resolution of 224.

## 6.6 Semantic Keypoint Matching

In Figure 10, we show additional examples of real image pairs from each of the 18 object categories in SPair-71k and our method's predicted correspondences. Our method is able to handle a variety of difficult cases such as large viewpoint transformations (e.g., the side and front views of the cow or aeroplane) and occlusions from other objects (e.g., the people on top of the motorbike or bars in front of the potted plant).

In Figure 11, we show additional examples of synthetic image pairs and our method's predicted correspondences. Many of the prompts were inspired by objects and compositions from PartiPrompts [51]. In the same setting as Section 4.3, we transfer the aggregation network tuned on real images to make these predictions. Our method is able to produce high-quality correspondences for these out-of-domain synthetic images, such as the astronaut riding a horse or raccoon playing chess.

## 6.7 Dense Warping

Our aggregation network, which was trained with sparse semantic keypoints as supervision, can also be used for dense warping. In Figure 12 we demonstrate how our dense nearest neighbor matches can be used to splice the appearance and structure of two images. Simply by bilinearly upsampling our Diffusion Hyperfeatures to the dimension of the input images and copying the nearest source pixel for every target pixel (i.e., a backward warp), our method is able to preserve fine-grained structures and textures such as the hair on the dog's face (right column, fourth row) or the texture of the cat's ear (right column, first row). In Figure 13 we show a similar visualization for synthetic images. In Figure 14 we also use these dense matches between the first frame and other frames of a video to propagate an object mask or semantic edit. Again, our method is able to preserve fine-grained textures such as the red spots on the cow or lines on the bear's vest (third row).

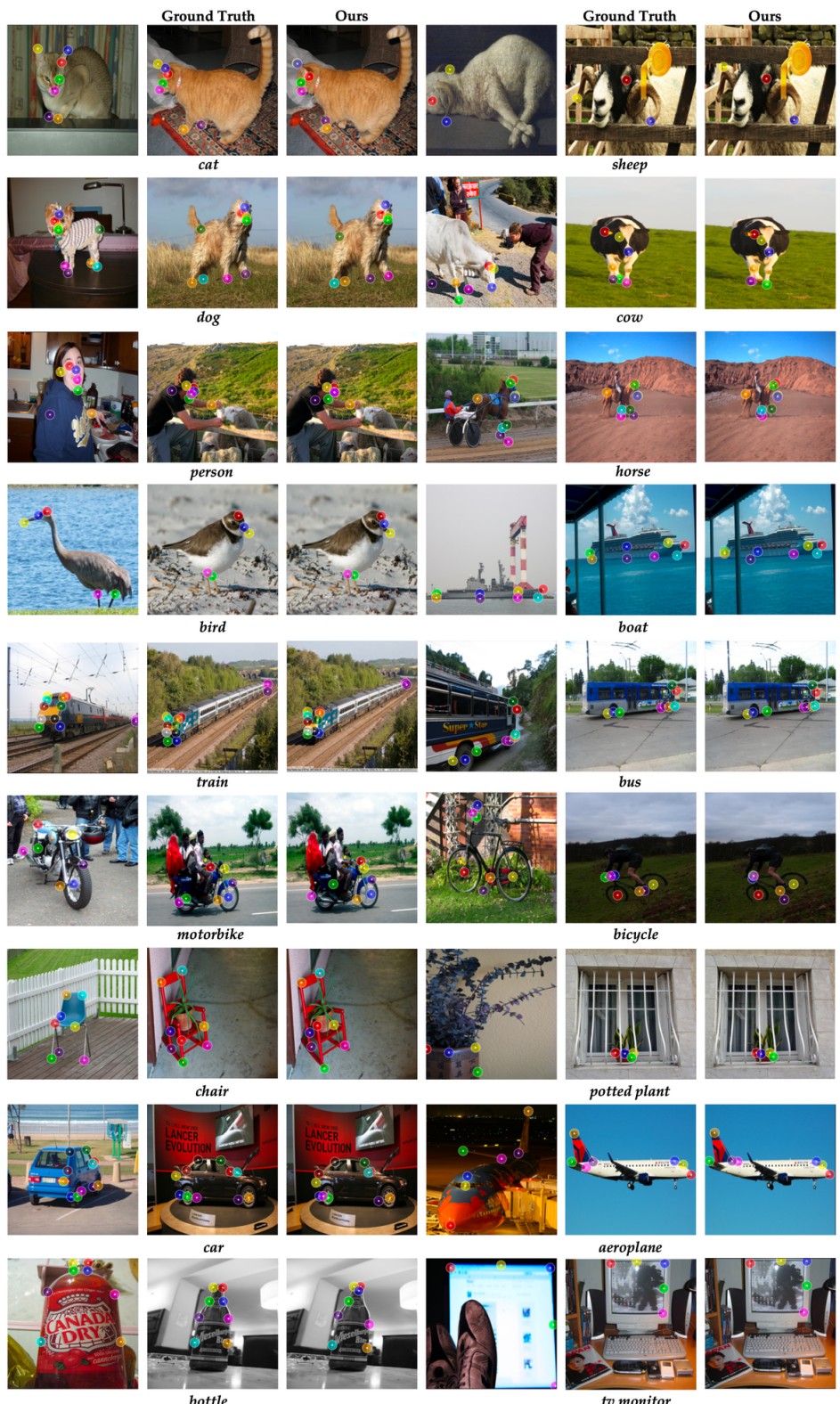

Figure 10: Additional examples of predicted correspondences from our method on real images from each of the 18 categories in SPair-71k.

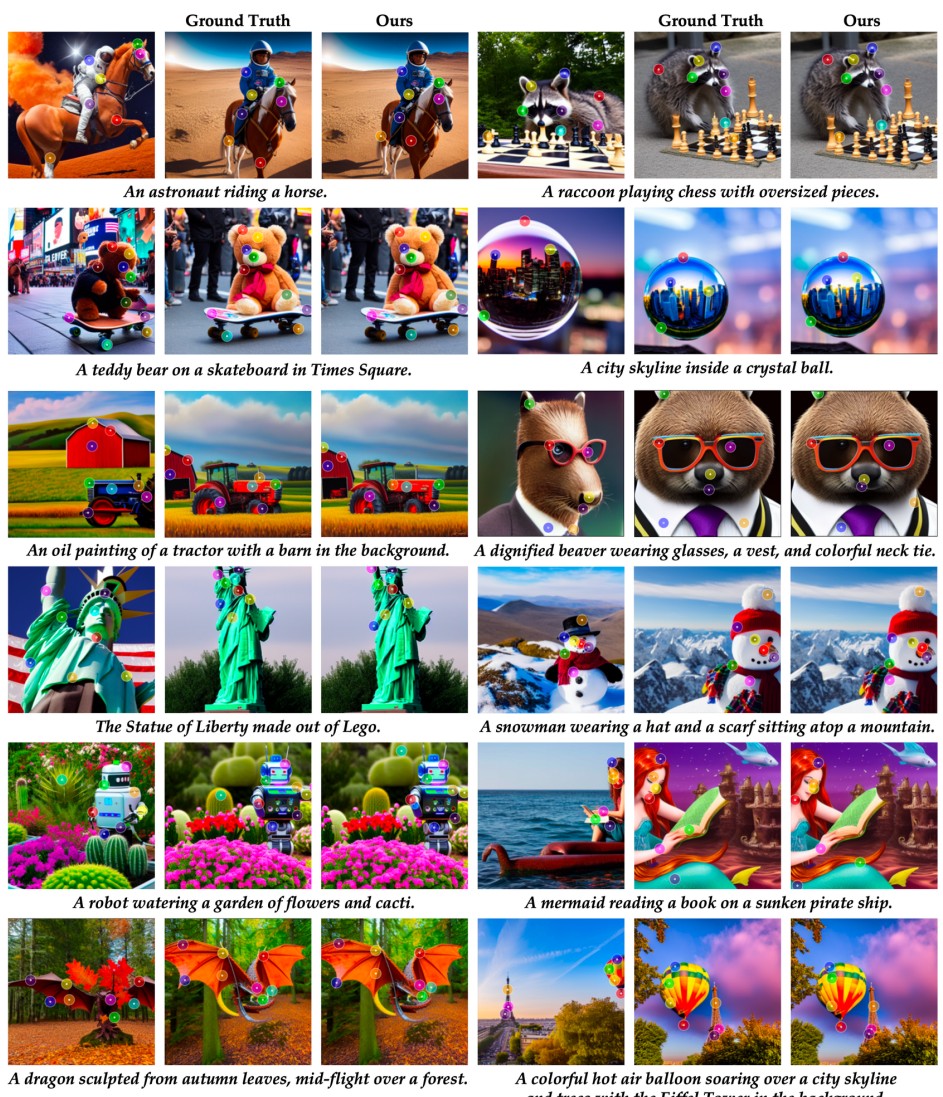

Figure 11: Additional examples of predicted correspondences from our method on synthetic images from a diverse set of prompts. Note that for synthetic images we transfer the aggregation network tuned on real images from SPair-71k.

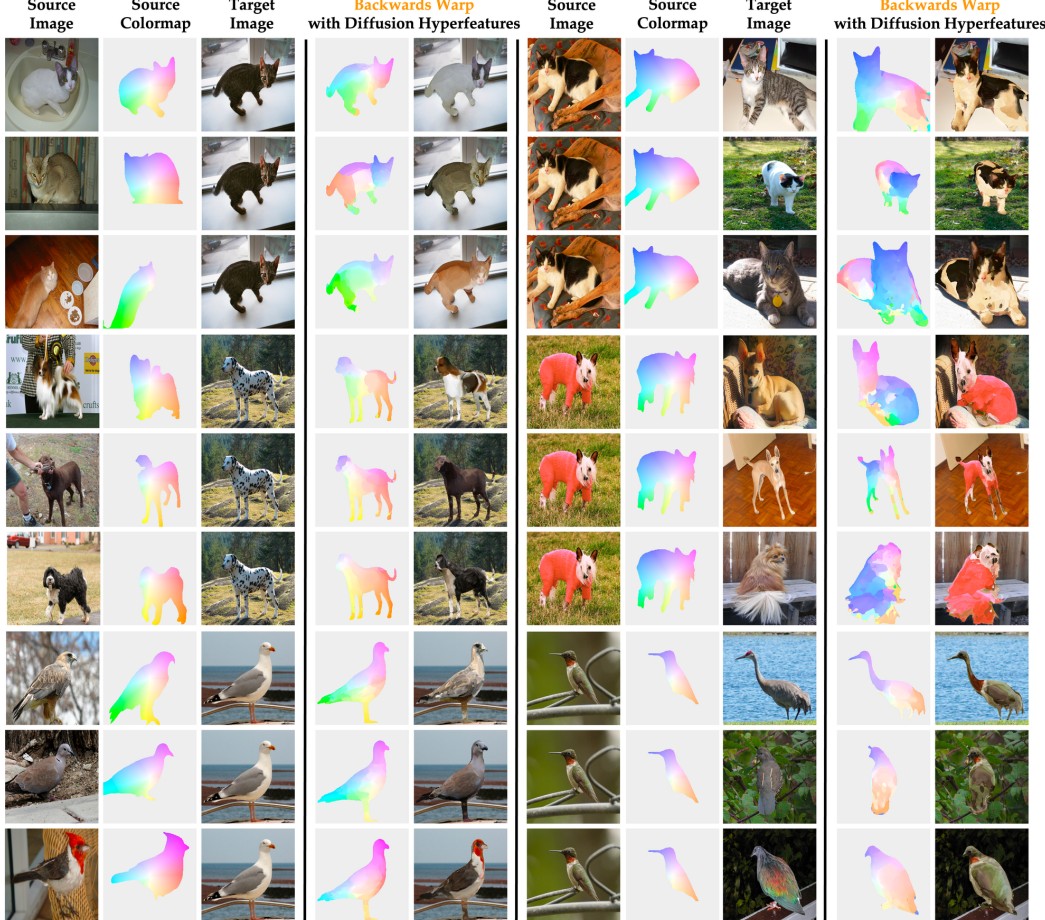

Figure 12: We show examples of dense backward warps using our method for real images from SPair-71k. We compute the nearest neighbors matches between the source and target image, constrained by an object mask [48], and warp either the source image or colormap accordingly.

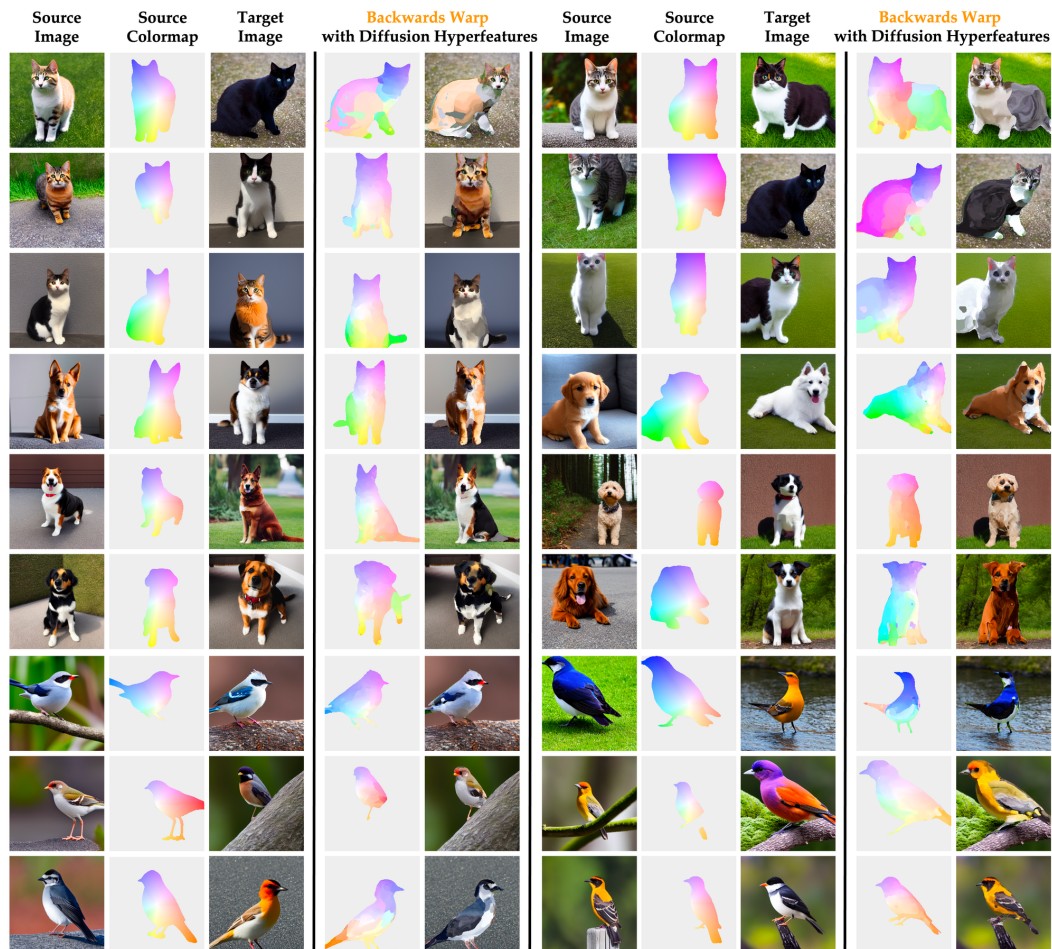

Figure 13: We show examples of dense backward warps using our method for synthetic images using the same procedure as the real images.

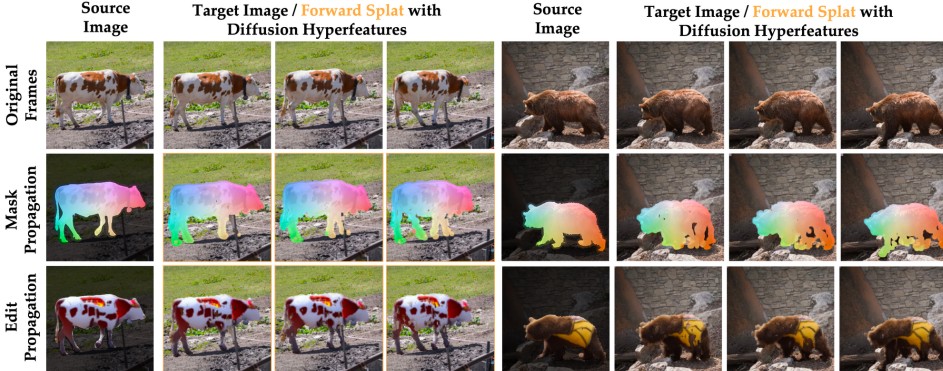

Figure 14: We show examples of dense forward splats using our method for real frames from DAVIS videos [35]. We compute the nearest neighbors matches between the first frame and later frames in the video (t=10, 20, 30), with no constraints, and propagate either a colormap or object edit [33] applied only to the first frame.