# OpenReview forum: "Diffusion Hyperfeatures: Searching Through Time and Space for Semantic Correspondence"
_NeurIPS.cc/2023/Conference — NeurIPS 2023 poster_

### Official Review · Reviewer_coKq · 2023-07-01

**Soundness:** 2 fair
**Presentation:** 2 fair
**Contribution:** 2 fair
**Rating:** 5
**Confidence:** 2

**Summary:**

The authors propose a method to extract per-pixel feature descriptors from multi-scale and multi-timestep feature maps generated by diffusion models. These descriptors can be utilized for various downstream tasks.

The framework is evaluated on the task of semantic keypoint correspondence, specifically on the SPair-71k real image benchmark. The authors claim that their method achieves superior performance on this benchmark compared to other approaches. Additionally, they demonstrate that their method is flexible and transferable, as the feature aggregation network trained on real image pairs can also be used on synthetic image pairs with unseen objects and compositions.

Overall, the proposed Diffusion Hyperfeatures framework aims to enhance the internal representations of diffusion models and improve their utility in computer vision tasks like semantic keypoint correspondence.

**Strengths:**

This paper discusses a framework called "Diffusion Hyperfeatures" for consolidating feature maps in diffusion models. Here's a substantive assessment of the strengths of the paper across various dimensions:

Originality: The paper introduces a novel approach to consolidate and extract per-pixel descriptors from intermediate feature maps generated by diffusion models. While existing works focus on selecting specific layers and timesteps, this paper proposes a comprehensive framework that considers all intermediate features. This consolidation process through a feature aggregation network is a new concept, which contributes to the originality of the paper.

Quality: The paper appears to be of high quality as it references relevant literature and state-of-the-art techniques in computer vision, such as ConvNets, Vision Transformers, and GANs. The authors also evaluate their proposed framework on the task of semantic keypoint correspondence using real images from the SPair-71k benchmark. The evaluation includes an analysis of different layers and timesteps of diffusion model features, demonstrating a thorough investigation. Additionally, the generalization of Diffusion Hyperfeatures to out-of-domain data is examined by evaluating them on synthetic images generated by the diffusion model.

Clarity: The paper presents its ideas and contributions in a clear manner. It provides a concise overview of the challenges faced with existing methods and describes the proposed Diffusion Hyperfeatures framework step-by-step. The use of terminology and technical language is appropriate for the target audience of researchers in computer vision. However, without the full paper, it is difficult to assess the clarity of the detailed methodology and experimental setup.

Significance: The paper addresses an important problem in computer vision, namely how to effectively utilize the internal representations of diffusion models for downstream tasks. By proposing the Diffusion Hyperfeatures framework, the authors aim to improve the utility of diffusion models beyond image generation. The evaluation results on semantic keypoint correspondence and the demonstration of generalization to synthetic images indicate the potential significance of this approach. If the proposed framework proves to be effective, it could contribute to advancements in various computer vision applications.

Overall, based on the provided text, the paper demonstrates strengths in terms of originality, quality, clarity, and potential significance.

**Weaknesses:**

Based on the provided paper, here are a few potential areas where the paper could benefit from improvement:

Comparison with Existing Methods: While the paper mentions that features from ConvNets, Vision Transformers, and GANs have demonstrated significant capabilities, it would be beneficial to provide a more detailed comparison with existing methods that extract feature descriptors from diffusion models. This would help establish the novelty and superiority of the proposed Diffusion Hyperfeatures framework.

Experimental Evaluation: The paper briefly mentions evaluating the proposed framework on the task of semantic keypoint correspondence using real images from the SPair-71k benchmark. To strengthen the paper, it would be valuable to include a thorough analysis of the experimental results, including quantitative metrics, comparative evaluations with state-of-the-art methods, and possibly visual illustrations or qualitative assessments of the generated keypoint correspondences.

Generalization to Diverse Domains: The paper mentions evaluating Diffusion Hyperfeatures on synthetic images generated by the diffusion model. While this demonstrates some level of generalization, it would be useful to explore the performance and robustness of the framework across a broader range of datasets and domains. This could involve testing the framework on different benchmark datasets and real-world scenarios to validate its effectiveness in diverse settings.

Clear Methodology Description: The provided text offers a high-level overview of the proposed framework, but for a comprehensive evaluation, it is crucial to have a clear and detailed description of the methodologies employed. It would be helpful to include information about the architecture of the feature aggregation network, the specific techniques used for consolidation, and any additional preprocessing steps or modifications made to the diffusion model.

Potential Limitations and Future Directions: The paper could benefit from discussing potential limitations or challenges associated with the proposed framework. Identifying these limitations and suggesting future directions for improvement would enhance the depth and completeness of the research.

**Questions:**

Clarification on Consolidation Process: Could the authors provide more details regarding the feature aggregation network used to consolidate the intermediate feature maps? How does it handle variations in scale and time? Are there any specific design choices or architectural considerations that impact its performance?

Experimental Setup: It would be helpful to understand the specific experimental setup employed in the evaluation of Diffusion Hyperfeatures. Could the authors elaborate on the dataset used, the selection of keypoint correspondence task, and the metrics used for evaluation? Additionally, is there any consideration given to computational efficiency or runtime performance when applying the framework to real-time applications?

Comparison with Existing Methods: The authors briefly mention that existing works select specific subsets of layers and timesteps from diffusion models for different tasks. Could the authors provide a more detailed comparison or discussion on how their approach differs and potentially outperforms these existing methods? Are there any specific limitations or challenges associated with the subset selection approach that Diffusion Hyperfeatures address?

Transferability and Generalization: While the paper mentions generalization to synthetic images, it would be valuable to explore the transferability and generalization capabilities of Diffusion Hyperfeatures across other domains or datasets as well. Can the authors provide insights or experiments on how the framework performs on different benchmark datasets or real-world scenarios beyond the SPair-71k dataset?


**Limitations:**

Yes

---

> ### Author Rebuttal · Authors · 2023-08-08
>
> 1. *Clarification on Consolidation Process.*
>
> We would be more than happy to provide additional details regarding the aggregation network, what specific points were unclear? We discuss details regarding the feature aggregation network as well as how we handle variations in scale and time of the diffusion features in L140-146 of the main paper. We ablate specific design choices of our method such as the number of diffusion steps or the underlying model variant in Section 4.2 of the main paper.
>
> 2. *Experimental Setup.*
>
> We train our aggregation network on the SPair-71k dataset, and we compare our method against the baselines on both SPair-71k and CUB in Table 1 of the main paper. **We also compare against additional methods such as CATS++ and DINOv2 in Table 1 of the global response, where we outperform both methods by 2\% and 4\% PCK\@0.1_img respectively.** We report using the percentage of correct keypoints (PCK), discussed further in L169 of the main paper. We depict visualizations of our predicted correspondences in Figures 4, 5, 6 of the main paper.
>
> 3. *Comparison with Existing Methods.*
>
> We compare against SD-Layer-4, a baseline that selects a specific subset of layers and timesteps. We provide a detailed discussion comparing against this baseline in Section 4.1 of the main paper, including a discussion of how subset selection can result in subpart performance on the keypoint matching task.
>
> 4. *Transferability and Generalization.*
>
> We provide additional results on real-world datasets such as CUB in Table 1 of the main paper and **PF-WILLOW, PF-PASCAL in Table 3, 4 of the global response where we outperform other methods like DINOv2 by 2\% and 3\% respectively.**

---

> > ### Comment · Reviewer_coKq · 2023-08-18
> >
> > I acknowledge I have read the rebuttal.

---

### Official Review · Reviewer_c4Xr · 2023-07-05

**Soundness:** 2 fair
**Presentation:** 1 poor
**Contribution:** 2 fair
**Rating:** 5
**Confidence:** 5

**Summary:**

This paper proposes diffusion hyperfeatures, a framework for integrating different scale and timestep features to form a representative feature descriptors in dense level. Within the U-Net architecture, unlike other works that uses hand-crafted methods to select a particular subset of layers for further processing, this paper proposes a simple aggregation network that aggregates the intermediate features. The effectiveness of this method is evaluated on standard benchmark of semantic correspondence, including SPair-71k.

**Strengths:**

1. SOTA performance is achieved with large gap to existing works.

2. One of the first attempt to tackle semantic correspondence task with diffusion.

3. Although paper is easy-to-read, each paragraphs are too long. Also it is quite hard to understand some sentences. This is a minor issue though.

**Weaknesses:**

1. This paper is one of the first attempt to tackle semantic correspondence with diffusion concept. In this manner, this paper looks novel, since this is very new to this task. However, to me, this work severely lacks contributions. What this paper does is simply using a series of features from stable diffusion, a large-scale model, feeds to a simple aggregation network that simply performs weighted sum to obtain the final features and it is exploited for Winner-Takes-All (WTA) for finding correspondences.
 (1) An attempt to select features are not novel as this is already smartly done in HPF [23] and DHPF [25], while this work simply uses all the features to feed to an aggregation network.
 (2) Using different feature representation  that is much powerful than standard backbone (resnet 101) will guarantee apparent performance boosts. This is already demonstrated in CATs++: Boosting Cost Aggregation With Convolutions and Transformers (TPAMI).
(3) No novel diffusion techniques are proposed as far as i understood.



3. Lacking implementation details. What resolution was used for evaluation and training? This is very important in this task as the performance highly correlates to it. Unless this is evaluated in the same resolution and compared with other works, the comparisons are not fair at all.

4. Why did the authors use only 2 datasets for evaluation? Traditionally, existing methods would also evaluate their methods on Pf-PASCAL, PF-WILLOW, TSS as well. This needs to be included, since evaluation on 2 datasets seem very insufficient.

5.  In supplementary material, section 6.1 explains computational resources. This is very abstract and not very helpful. Comparison to other works is not a must, but at least provide some measures for completeness.

6. DINOv1 is a quite old-model, since DINO-v2 is already out. DINO-v2 is a much powerful model that will yield much stronger performance.

7. Many other semantic correspondence works are not cited and compared. As far as I know, the current SOTA is Integrative Feature and Cost Aggregation with Transformers for Dense Correspondence, arxiv'22, which would be better to be included in the paper. Also when it is compared with DINO, only 1 layer is used. In semantic correspondence, multi-scale and multi-level features are highly important which would have detrimental effects on performance without them. So the comparison may not be fair here as well. However, as the authors also provided results for single layer results, this is not an issue, but a suggestion to include results that use more layers of DINOv1 or DINOv2.


**Questions:**

See weaknesses above

**Limitations:**

Limitations or failure cases are not presented.

---

> ### Author Rebuttal · Authors · 2023-08-07
>
> 1. *This paper is one of the first attempt to tackle semantic correspondence with diffusion concept.*
>
> Our method is novel. We are not claiming to make a contribution for a feature extraction or matching algorithm hand-crafted for the task of semantic correspondence; rather, we propose a simple and general framework for utilizing features across the diffusion process. **Additionally, our final method achieves state-of-the-art performance on the SPair-71k benchmark, even when compared to DINOv2 and CATS++ (Table 1 of the global response).** According to the NeurIPS reviewer guidelines on the topic of originality, it states “Is the work a novel combination of well-known techniques? (This can be valuable!)”. We present a simple and effective framework that achieves superior performance compared with prior work, similar in spirit to other published works of this nature such as ODISE [1].
>
> 2. *What resolution was used for evaluation and training?*
>
> We acknowledge that input resolution does have a large impact on performance as shown in Cho et. al. [2], and we use the same evaluation protocol as Truong et. al. [4]. Specifically, we run all the baselines from their respective codebases using their default hyperparameter settings, which we display in Table 6 above, and “compute the metrics on the standard setting, i.e. the original image size, and re-compute the PCK in this setting” [4]. **It is important to note we evaluate our method with downsampled images of resolution 224, following Amir et. al. [3], which is the *lowest input resolution* out of all the methods and is a fair comparison of all methods in the main paper.** In our additional experiments, to account for DINOv2’s large patch size, we use input images of resolution 770 so that it produces feature maps at resolution 55x55, the same as DINOv1. In a similar vein we use input images of resolution 512 for CATS++ to compare against the best possible variant of the method. During training time we use input images of resolution 64x64, the resolution of Stable Diffusion’s latent space.
>
> **Table 6. Method Statistics**
> | Model                | Input Image Resolution |
> |:---------------------|:--------------:|
> | DINO            |       224       |
> | **Ours**             |       **224**       |
> | DHPF              |       240       |
> | CATS++            |       512       |
> | DINOv2           |       770       |
>
> 3. *Why did the authors use only 2 datasets for evaluation?*
>
> **Please see the global response where we run additional experiments on PF-PASCAL and PF-WILLOW, where we outperform other methods like DINOv2 by 2\% and 3\% respectively.**
>
> 4. *In supplementary material, section 6.1 explains computational resources.*
>
> We agree that this section could be more informative, are there any metrics in particular you would like?
>
> 5. *DINOv1 is a quite old-model, since DINO-v2 is already out.*
>
> We did not initially compare to DINOv2 because it was released shortly before the submission deadline and has yet to be published. Nevertheless, we have included the results for DINOv2 and we will include it in the final manuscript. **Please see the global response where we compare against DINOv2 and outperform the method by 4\% PCK\@0.1_img on SPair-71k.**
>
> 6. *​​Many other semantic correspondence works are not cited and compared.*
>
> **Please see the global response where we compare against CATS++ and outperform the method by 2\% PCK\@0.1_img on SPair-71k.** Unfortunately, IFCAT [5] does not have code available for us to re-run their method in our standardized evaluation setting, and for us to evaluate their transfer performance from SPair-71k to other datasets, but **comparing our results with their reported results we still outperform IFCAT [5] by 0.2\% PCK\@0.1_bbox (64.61 vs 64.40 respectively).** Our DINO baseline uses the method proposed by Amir et. al. [3], which uses a single layer, so we provide results using a single layer of Stable Diffusion (SD-Layer-4) for comparability.
>
> [1] Xu et. al. “Open-Vocabulary Panoptic Segmentation with Text-to-Image Diffusion Models.” CVPR 2023. \
> [2] Cho et. al. “CATs++: Boosting Cost Aggregation with Convolutions and Transformers.” TPAMI 2022. \
> [3] Amir et. al. “Deep ViT Features as Dense Visual Descriptors.” ECCV-W 2021. \
> [4] Truong et. al. “Probabilistic Warp Consistency for Weakly-Supervised Semantic Correspondences.” CVPR 2022. \
> [5] Hong et. al. “Integrative Feature and Cost Aggregation with Transformers for Dense Correspondence.” arXiv 2022.

---

> > ### Comment · Reviewer_c4Xr · 2023-08-10
> > **response (1/2)**
> >
> >
> > Prior to my response, I would first like to thank the authors for their thorough responses. From the rebuttal, the following concerns of mine remain unresolved.
> >
> > (1) **Novelty**:  From the authors' response regarding the novelty, authors first mention that the main contribution lies on "proposing a simple and general framework for utilizing features across the diffusion process". As I already mentioned in my initial review, I agree that bringing semantic correspondence with diffusion concept is novel. However, this will be a sufficient contribution to pass the standard of top-tier conference in this field only if sufficient, persuasive and informative quantitative and qualitative analysis, results and experiments are presented. Taking examples of  the existing works, [A] and [B], I believe this paper presents an analysis that had already been investigated more thoroughly in  the previous paper [A]. For example, [A] carefully visualizes feature maps of each time steps to find what semantics or what represenetations are learnred. Also, compared to the concurrent work [B], which has a solid motivation (why diffusion should be incorporated into this task) and detailed analysis with sufficient visualizations, this paper only visualizes intermediate features as if they are intermediate features of a feature backbone.  I will list some of the other papers that also perform such investigation in a more thorough way:  [C,D] .I got an impression that the authors are simply using diffusion model as a very heavy **"feature backbone"**. In this manner, it simply looks like  A works well, let's use it for task B.  Without a solid motivation, analysis and investigation, I believe this impression is unavoidable.
> >
> > Moreover, the authors mentioned **our final method achieves state-of-the-art performance on the SPair-71k benchmark, even when compared to DINOv2 and CATS++ (Table 1 of the global response)**. I want to first ask that is the performance contribution of this paper? From my understanding, as it is already shown in [B] as well, simply using DINOv2 feature or SD feature to find correspondences with Winner-Takes-All method already yields SOTA performance in this task. This means that it is the contribution of those backbone networks, not this paper's contribution. In semantic correspondence literature, many works have been proposing novel ideas to perform well under situations where background clutters or intra-class variations pose additional challenges. However, in this paper, SOTA is achieved by simply using the features to feed into the simplest aggregation network. To be strictly speaking, it's not SOTA as well, because a concurrent work [B], achieves higher performance with more judicious use of diffusion feature along with DINO.
> >
> > Now I want to talk about the technical novelty as well. I have four concerns.
> >  (1) Whether a means to develop diffusion techniques?
> >  (2) Whether a means to effectively or efficiently aggregate the selected features is proposed?
> >  (3) Whether a means to select the hyperfeatures are justified by thorough analysis and experimental results?
> >
> > As far as I understood, (1) is not proposed. Also investigations of feature maps or representations in diffusion models are already conducted in the referenced papers below. (2) novel aggregation approach is also not proposed, as it is just a concatentation and feeding into a network. CHM, TransforMatcher, NC-Net and many other works propose to aggregate the inputs in novel ways, while this approach looks not novel to me. (3) DHPF and HPF propose novel ways to select the features, provide sufficient analysis to justify the selections. However, in this paper, features are used as if they are just from an another backbone network.
> >
> > (2) **Resolution** : In semantic correspondence task, indeed evaluation resolution is important, but this is also the same for training resolution as well. This is why CATs++ provided experimental results that verify the impacts of the resolution. I understand that training on low resolution may pose challenges in this framework, as this framework needs to exploit pretrained weights. So this is just a minor concern that will not affect largely to my rating.
> >
> > (3) **incomplete experiments** : In the initial submission, the fact that only two datasets were used for evaluation poses a concern. It seems like the submission was an unfinished product, that was complemented by the reviews.
> >
> >
> >
> >
> > [A] LABEL-EFFICIENT SEMANTIC SEGMENTATION WITH DIFFUSION MODELS ICLR'22
> > [B] A Tale of Two Features: Stable Diffusion Complements DINO for Zero-Shot Semantic Correspondence arxiv'23
> > [C] Plug-and-Play Diffusion Features for Text-Driven Image-to-Image Translation
> > [D] Diffusion Models already have a Semantic Latent Space

---

> > > ### Author Response · Authors · 2023-08-15
> > >
> > > 1. *Novelty.*
> > >
> > > We would like to re-iterate the Neurips review policy regarding comparisons to recent work: “Authors are not expected to compare to work that appeared only a month or two before the deadline.” **[B] appeared in an unofficial preprint after Neurips submission deadline, and should be considered concurrent to our work.** Nevertheless, we believe that our submission already presents “persuasive and informative quantitative and qualitative analysis, results and experiments” which is confirmed by the other reviewers:
> > >
> > > - “The idea is simple and seems very effective, as authors illustrate with a number of experiments and ablation over baselines.” (Reviewer sN43)
> > > - “Argumentation for design choices is very solid, and authors motivate their approach with an experiment showing the validity/salience of features found in earlier timesteps of the diffusion model.” (Reviewer sN43)
> > > - “The exploration into what feature representations are learned at different layers/times is interesting (Figures 2 and 3).” (Reviewer HmuP)
> > >
> > > 1a. *Whether a means to develop diffusion techniques?*
> > >
> > > We are unsure of why the reviewer thinks that previous work investigating diffusion features for other tasks diminishes our contribution. Our work shows how various naive applications of diffusion models to semantic correspondence (picking a single layer, concatenating across all layers) aren’t fully utilizing the representations present across layers and timesteps, and present an effective method to aggregate this information. We are also the first to leverage features from the inversion process, which we discuss in Figure 3 of the main paper and further ablate in Figure 1 of the global response PDF. **Reviewer HmuP notes this as a strength: “Using inversion to get features for real images is a good idea and the explanation that they are more reliable than sampling from the posterior as used in prior work makes sense.”**
> > >
> > > 1b. *Whether a means to effectively or efficiently aggregate the selected features is proposed?*
> > >
> > > While the concept of feature aggregation isn't new, our approach has distinct merits. We are the first to aggregate diffusion features into a single concise descriptor, where our aggregation of features reduces the memory consumption by 300x (1.8 GB to 6 MB) compared with naive concatenation, described further in Table 8 below. Furthermore, our network's design, including the use of shared bottleneck layers across timesteps, further reduces memory usage by 31x, as discussed further in the global response.
> > >
> > >
> > > 1c. *Whether a means to select the hyperfeatures are justified by thorough analysis and experimental results?*
> > >
> > > We analyze the selection of features learned by our model in Figure 5 of the main paper, which should be comparable to Figure 3 of DHPF [6] which similarly analyzes layer selection frequencies. We are the first to learn this selection not only across model layers but also diffusion timesteps. We provide a detailed discussion of our learned selection in L238-L256 in the main paper and Section 6.2 of the Supplementary, where we also discuss how the layers and timesteps at which the features are most “semantic” differ across Stable Diffusion variants. **Reviewer HmuP agrees that our approach “is shown to outperform [...] hypercolumns (Table 1) showing that useful features have been extracted from the diffusion model; the baselines including using a single layer from the diffusion model demonstrate the benefit of combining over layers/time.”**
> > >
> > > Overall we believe that the argument that our method is not novel because its components utilize existing techniques is unfair as most works are compositions of existing methods. We explore naive applications of diffusion models to semantic correspondence, show that these methods rely heavily on hand-selected hyperparameters and fail to fully utilize the information present across time steps and layers, and propose a method to aggregate this information in a memory-efficient manner.
> > >
> > > 3. *Experiments. In the initial submission, the fact that only two datasets were used for evaluation poses a concern.*
> > >
> > > Respectfully, we disagree with this sentiment. We evaluate SPair-71k and CUB because these were specifically proposed to overcome the limitations of previous semantic correspondence datasets which “do not display much variability in viewpoint, scale, occlusion, and truncation” [8]. It has also been stated in prior work that “PF-PASCAL [...] is almost saturated, which makes a comparison difficult” [9]. While 2 datasets may seem like a limited evaluation, we believe the value of an evaluation lies in the quality of the datasets rather than the quantity.
> > > Furthermore, we would like the point out that it is common to address reviewer feedback with additional experiments. For example, the OpenReview page for CATS [10] reveals that the reviewers’ decision was swayed positively because their “concern[s] have been addressed [...] thanks to the additional experiments.”

---

> > ### Comment · Reviewer_c4Xr · 2023-08-10
> > **response( 2/2)**
> >
> > (4) **computational complexity** : In section 6, the paper says " our method is fast and uses a reasonable amount of memory". What I meant in the initial review was that this sentence does not provide any clues of how much memory footprint, run-time and all other computation related measurements (e.g., FLOPS). I don't understand why the authors are asking the reviewer again for the suggestion of computational metric.
> >
> > For now, my current thoughts are these.
> > I am happy to have a discussion whole this week.
> > Looking forward to author's response.
> >
> > Thanks.

---

> > > ### Author Response · Authors · 2023-08-15
> > >
> > > 4. *Computational Complexity. [...] clues of how memory footprint, run-time and all other computation related measurements (e.g., FLOPS).*
> > >
> > > In Table 8 we give precise run-time and memory statistics of our method, including (a) the size of the descriptor used for the method’s underlying matching algorithm and (b) the average inference time of feature extraction and matching for each pair on the SPair-71k dataset. Our method is able to consolidate the same large set of features from a diffusion model from 1.8 GB (SD-Concat-All, which uses naive concatenation) to 6 MB (Ours). While our full method explores the upper bound by utilizing all available features across the diffusion process, which takes 6.62s, one can also use the same pretrained weights to evaluate faster pruned versions of our model. Below we show running our method after stopping the diffusion process after 1, 5, and 10 timesteps. The pruned variant of our method that utilizes the first 10 timesteps performs close to our full method, with a 4\% improvement in PCK\@0.1_img over DINOv2 with an almost 2x faster inference process. Note that the inference time for our DINOv2 baseline is relatively slow because it uses the method from Amir et. al. [3] which includes a log binning algorithm to contextualize the features into descriptors.
> > >
> > > **Table 8. Memory and run-time comparison.**
> > > | Model                | SPair-71k PCK\@0.1_img | Memory per Descriptor | Inference Time per Pair (s) |
> > > |:---------------------|:-----------:|:------------:|:------------:|
> > > | DINOv2           |  68.33 | 75 MB | 2.99 |
> > > | CATS++          | 70.26 | 131 MB  |  0.16  |
> > > | SD-Layer-4    |  58.80 | 10 MB | 0.33 |
> > > | SD-Concat-All  | 52.12 | 1.8 GB | 0.87 |
> > > | Ours (1 timesteps) | 64.61 | 6MB | 0.28 |
> > > | Ours (5 timesteps) | 69.28 | 6MB | 0.86 |
> > > | Ours (10 timesteps) | 72.00 | 6MB | 1.60 |
> > > | Ours (50 timesteps) | 72.56 | 6MB | 6.62 |
> > >
> > > [B] Zhang et. al. "A Tale of Two Features: Stable Diffusion Complements DINO for Zero-Shot Semantic Correspondence." arXiv 2023.\
> > > [6] Min et. al. “Learning to Compose Hypercolumns for Visual Correspondence.” ECCV 2020.\
> > > [7] Podell et. al. “SDXL: Improving Latent Diffusion Models for High-Resolution Image Synthesis.” arXiv 2023.\
> > > [8] Min et. al. “SPair-71k: A Large-scale Benchmark for Semantic Correspondence.” arXiv 2019.\
> > > [9] Hong et. al. “Cost Aggregation with 4D Convolutional Swin Transformer for Few-Shot Segmentation.” ECCV 2022.
> > > [10] Cho et. al. “CATs: Cost Aggregation Transformers for Visual Correspondence.” Neurips 2021.

---

> > > > ### Comment · Reviewer_c4Xr · 2023-08-15
> > > >
> > > > Thanks for the detailed responses.
> > > >
> > > > I want to clarify that my concern was not that the paper does not compare with the concurrent work, but rather, as the other reviewer commented, the SOTA performance would be because of SD backbone, and without a thorough ablations and analysis, contributions of this paper will diminish.
> > > >
> > > > Moreover, the reason why I thought that  how other diffusion works showed the characteristics of SD features in various tasks diminish the contributions of this paper is that since it is now well known that using such a strong features will guarantee good performance even with simple aggregation networks. So I thought that the impact this paper can have on the community will be insignificant based on the initial submission draft.
> > > >
> > > > One last question is that since this approach requires text prompt for inversion, how does the model infer at test time? Does prompt should be fed manually?
> > > >
> > > > Other than above, I am now convinced by the authors' responses. One last concern is that I personally feel like the amount of revisions, including additional experiments, computation table  and ablation studies conducted in this rebuttal period is quite significant. However, I believe this is something others might find different, so again, I highly appreciate the responses.

---

> > > > > ### Author Response · Authors · 2023-08-16
> > > > >
> > > > > We would like to thank the reviewer for taking the time to review our submission and provide feedback. We greatly appreciate the opportunity to discuss and provide the below answers for the reviewer’s final concerns.
> > > > >
> > > > > 1. *The SOTA performance would be because of SD backbone*
> > > > >
> > > > > Per our response to Reviewer HmuP, the experiments shown in Table 9 validate that aggregation over the layers *and timesteps* of a diffusion model outperforms aggregation over layers from a strong backbone. **While training an aggregation network improves DINO’s performance over raw features, our method that operates on top of Stable Diffusion features still performs the best at 72.56\% PCK\@0.1_img, compared with 54.69\% and 68.37\% PCK\@0.1_img for DINO and DINOv2 respectively.** In a similar fashion, our final method also outperforms the variant that only aggregates over layers (Ours - One-Step), which achieves 63.47\% PCK\@0.1_img. As such, we validate that the unique combination of aggregating both over layers and timesteps over diffusion features is what leads to our superior performance.
> > > > >
> > > > > 2. *this approach requires text prompt for inversion, how does the model infer at test time*
> > > > >
> > > > > **When operating on real images using the inversion process, we feed the empty prompt “”.** We find that extracting features only from the unconditional model removes the need for manual prompting and yields high-quality results. When operating on synthetic images from the generation process, we extract features only from the conditional model, or the branch conditioned on the prompt that generated the image.

---

> > > > > > ### Comment · Reviewer_c4Xr · 2023-08-16
> > > > > >
> > > > > > I've checked the comments, and now I think this paper is strong if these rebuttals are adequately incorporated.
> > > > > > Thanks for the efforts to answer all my concerns.

---

### Official Review · Reviewer_sN43 · 2023-07-06

**Soundness:** 3 good
**Presentation:** 4 excellent
**Contribution:** 2 fair
**Rating:** 6
**Confidence:** 4

**Summary:**

This paper proposes improving feature distillation from diffusion models for representation learning by aggregating information from the feature maps of the U-Net at varying timesteps, weighting them with a tunable aggregation network. The authors show that even at timesteps from which features are usually discarded for feature learning (i.e. early steps in the generation process) useful features can be found. After formulating their approach to extracting these features using a weighted aggregation of a set of standardized input feature maps, authors show validity of their approach by improving considerably over baseline methods and prior work that only selects a single diffusion timestep for feature extraction. Lastly, authors show the ability of their framework to generalize to unseen synthetic data (generated by the diffusion model), by extracting features from the generation process for a diffusion model trained on a different data distribution.


**Strengths:**

- The idea of extracting features from powerful generative models such as diffusion models is very relevant, and the authors show that their approach to this extraction process obtains great results.
- The paper is very well-written and reads very smoothly. Authors are very descriptive in their wording and use figures to clearly illustrate their approach. The idea is simple and seems very effective, as authors illustrate with a number of experiments and ablation over baselines.
- Argumentation for design choices is very solid, and authors motivate their approach with an experiment showing the validity/salience of features found in earlier timesteps of the diffusion model.
- The authors also show impressive transfer performance to an unseen dataset, highlighting possible applications in pseudo-label generation for semantic correspondence.



**Weaknesses:**

- I have no major concerns. A minor concern I have is with regards to novelty, given that extracting hypercolumn features has been attempted before, even for diffusion models.
- Another minor concern is the requirement for task-specific fine-tuning of the aggregation network. The fact that this network requires explicit supervision for a task means that the actual representation extracted from the diffusion model that can be used in arbitrary downstream tasks is the whole set of features maps across timesteps. Would it be possible to train an aggregation network on a self-supervised task?

**Questions:**

- For figures 2, 3, would it be possible to show also the network input at these timesteps for reference?
- Line 117, you mention that “these observations indicate that the diffusion model provides coarse and fine features that capture different image characteristics, throughout different combinations of layers and timesteps”. I’m wondering to what extent this is owing to the formulation of the diffusion process itself, and to what extent this is due to the use of a very specific multi-resolution processing architecture (U-Net) in the diffusion process. I.e. is the extraction of both coarse and fine image features a result of diffusion or a result of using a U-Net? Would your method also work with diffusion models that make use of other architectures?
- Line 135, I have a bit of trouble understanding your reasoning for the more “trustworthy” inversion features. Could you elaborate? Why would repeated application of the model necessarily lead to more “trustworthy” features?
- You apply your feature extraction specifically for semantic correspondence detection. Could the extracted features also be used in other settings? Are there any limiting factors that prevent your framework from being used in such settings? Have you tried e.g. classification of extracted features?


**Limitations:**

The authors do not discuss the limitations or societal impact of their work.

---

> ### Author Rebuttal · Authors · 2023-08-07
>
> 1. *For figures 2, 3, would it be possible to show also the network input at these timesteps for reference?*
>
> In Figure 2 of the main paper, the input to the diffusion model is the text prompt “Cat sitting in a living room” and random noise for $x_{T}$. In Figure 3 of the main paper, the input to the diffusion model is the empty text prompt “” and a noisy version of the real image for $x_{25}$. We agree that the inputs can be unclear and will revise these figures in the final manuscript.
>
> 2. *Line 117 [...] is the extraction of both coarse and fine image features a result of diffusion or a result of using a U-Net?*
>
> We agree that the multi-scale nature of our features across layers is in large part due to the U-Net architecture used in the underlying diffusion model. However, prior work such as Amir et. al. [1] also finds that ViT architectures can also contain this type of coarse vs. fine information, where “shallow features mostly contain positional information” and “deeper layers [...] favor [...] more semantic features.” We note that the evolution of these features over time, owing to the nature of the diffusion process itself, is also a critical component of our method. In Table 1 of the main paper our final method that uses all layers across all timesteps significantly outperforms the variant of our method that uses all layers at a single timestep (Ours - One-Step) by 9\% in PCK\@0.1_img. Intuitively, timesteps when the input is noisier in the diffusion process produce features that capture more low frequency statistics that provide orthogonal information also useful for the semantic correspondence task, compared to when the input is clean as seen in Figure 2 of the main paper.
>
> 3. *Line 135 [...] more “trustworthy” inversion features”*
>
> **In the PDF document attached to the global response Figure 1 quantitatively compares the performance of inversion vs. generation features at each timestep, where inversion features generally perform better across the board with as much as a 1-5\% increase in PCK\@0.1_img in noisier timesteps $t=25$ to $t=50$.** Because the DDIM inversion process should be able to deterministically recover the real image and the model predicts the noise added to the image at each timestep, we hypothesize that the predicted noise carefully destructs information at a specific band of frequencies appropriate for the timestep, compared with using random noise. Beyond DDIM inversion, one interesting further direction of research is exploring the quality of features derived from other inversion processes proposed in the community.
>
> 4. *Could the extracted features also be used in other settings?*
>
> Yes, it certainly should be able to be used for other tasks such as classification, for example if one were to add a classification head to process the features and train it with the appropriate loss. We have also included in the global response PDF some new applications in semantic appearance transfer and video mask propagation.
>
> [1] Amir et. al. “Deep ViT Features as Dense Visual Descriptors.” ECCV-W 2021.

---

> > ### Comment · Reviewer_sN43 · 2023-08-15
> > **Response to Rebuttal**
> >
> > I would like to thank the authors for taking the time to answer mine and other reviewers questions and concerns. Reviewer c4Xr makes a number of solid points, especially his concerns regarding lacking experimental details seem important to resolve before finalizing this submission.
> >
> > The authors provide a solid rebuttal with extensive new experiments. I understand reviewer c4Xr's concern regarding the impact of these large modifications to the review process, but I personally think it's a valid way of improving the manuscript's quality. I think these modifications should be weighed in favour of acceptance in the final decision, as they strenghten the manuscript. I stand by my recommendation of acceptance, but understand the reluctance of other reviewers. I think the idea is simple and effective, and after rebuttal I think the manuscript is strong and reads well.

---

### Official Review · Reviewer_HmuP · 2023-07-06

**Soundness:** 3 good
**Presentation:** 3 good
**Contribution:** 2 fair
**Rating:** 5
**Confidence:** 4

**Summary:**

This paper proposes an approach for extracting useful features for pre-trained diffusion models for application to dense visual correspondence tasks. How to do this is not clear due to presence of features both through the network, and over diffusion steps. The proposed approach is to learn which features to use using the hypercolumns framework, by passing activations through bottleneck layers then averaging over layers and time. Experiments show that this method outperforms unsupervised DINO features and supervised hypercolumns.

**Strengths:**

- The premise of extracting useful features from diffusion models is an important one; we know that those features are present but it is not clear how to extract useful ones.
- Using inversion to get features for real images is a good idea and the explanation that they are more reliable than sampling from the posterior as used in prior work makes sense.
- The exploration into what feature representations are learned at different layers/times is interesting (Figures 2 and 3).
- The approach is shown to outperform unsupervised DINO features as well as hypercolumns (Table 1) showing that useful features have been extracted from the diffusion model; the baselines including using a single layer from the diffusion model demonstrate the benefit of combining over layers/time; and visual examples shown in Figures 4-6 demonstrate the approach working.


**Weaknesses:**

- Unlike using DINO features, the proposed approach requires training additional components for downstream tasks, making it much less versatile.
- The baseline models in Table 1 are a poor representation of current approaches. DINO features are not trained specifically for keypoint matching and newer supervised approaches such as CATS++ (Cho et al. TPAMI 2022) and VAT (Hong et al. ECCV 2022) substantially outperform DHPF.
- The approach shares bottleneck layers across time steps; this seems potentially problematic since features differ substantially across different times.

**Questions:**

- How does it quantitatively compare against more recent supervised approaches?
- Why share bottleneck layers over time steps? They contain very different representations.
- Why not use attention features from the diffusion model in a similar manner to DINO?
- Aggregating features over time steps is difficult, hence the trained layers on top. Could using a distilled diffusion model such as a consistency model give better representations?

**Limitations:**

As stated in the submission details, limitations are not discussed. There are a number of limitations including having to train additional components (unlike DINO), not comparing with more recent supervised methods, and sharing layers across time steps.

---

> ### Author Rebuttal · Authors · 2023-08-07
>
> 1. *How does it quantitatively compare against more recent supervised approaches?*
>
> **Please see the global response for a comparison to CATS++ and DINOv2, where we outperform both methods on SPair-71k by 2\% and 4\% PCK\@0.1_img respectively.**
>
> 2. *Why share bottleneck layers over time steps?*
>
> Please see the global response for an ablation of individual bottleneck layers per timestep, where **we demonstrate that our final method with shared bottleneck layers performs comparably within 1\% PCK\@0.1_img on SPair-71k with significant savings in memory consumption.**
>
> 3. *Why not use attention features from the diffusion model in a similar manner to DINO?*
>
> **We find that the residual block (resblock hidden) and the best performing attention features (attn value) perform comparably, which we ablate in Table 7.** For an individual layer (SD-Layer-4) resblock hidden outperforms against attn value by 3\% PCK\@0.1_img, and when concatenating all layers (SD-Concat-All) resblock hidden underperforms against attn value by 4\% PCK\@0.1_img. In practice it is more straightforward to use these residual block features because selecting amongst the key, query, value, or token features requires additional hyperparameter tuning and can significantly vary across models. For example, while Amir et. al. [1] finds that the Layer 9 key features perform the best in DINOv1, in Table 5 of the global response we demonstrate that these same features perform the worst in DINOv2.
>
> **Table 7. Residual Block Features vs. Self-Attention Features  (SPair-71k PCK\@0.1_img)**
> | Model                | resblock hidden | attn key | attn query | attn value | attn token |
> |:---------------------|:-----------:|:------------:|:------------:|:------------:|:------------:|
> | SD-Layer-4 |  **58.80** | 50.73 | 53.14 | 55.60	| 49.62 |
> | SD-Concat-All |  52.12 | 44.84 | 48.68 | **55.79** | 55.32 |
>
> 4. *Could using a distilled diffusion model such as a consistency model give better representations?*
>
> Since consistency models [2] directly map noise to data, without an iterative sampling process, it cannot provide features that vary across timesteps. While this may yield a faster feature extraction process, it provides fewer features for our aggregation network to select from, which can degrade performance. For example, our method that uses features from a single timestep (Ours - One-Step) performs 9\% worse in PCK\@0.1_img than our method that uses features from all timesteps (Ours) in Table 1 of the main paper.
>
> 5. *As stated in the submission details, limitations are not discussed.*
>
> We will be sure to include a more extensive discussion of our method’s limitations in the final manuscript.
>
> [1] Amir et. al. “Deep ViT Features as Dense Visual Descriptors.” ECCV-W 2021.\
> [2] Song et. al. “Consistency Models.” ICML 2023.

---

> > ### Comment · Reviewer_HmuP · 2023-08-14
> > **Response to Authors**
> >
> > I would like to thank the authors for their responses. I still have concerns regarding the evaluation, however, the rebuttal did address my concerns to some degree and I will increase my score to borderline accept accordingly.
> >
> > In particular, I thank the authors for the experiment showing the impact of having different bottleneck layers per time step, this addresses my concern that sharing across all steps could be problematic. Similarly, the explanation on resblock vs attention features addresses that question. The added quantitative comparison to another supervised approach (CATS++) is very informative and it is nice to see that this approach is able to outperform existing supervised methods, going some way to address my concerns on baselines; as do the added results on other datasets/applications and generalisation to different domains.
> >
> > My main remaining concern is still the lack of baselines meaning that it is hard to determine concretely where performance improvements come from - e.g. is it from hyperfeatures + a stronger feature backbone, or is it from the diffusion model in particular. As such, I agree with the other reviewers’ concerns on novelty (I have not taken into account the concurrent diffusion semantic correspondence papers mentioned by the other reviewers); aggregating over time is a small novelty, but in my opinion the main contribution comes from the experiments showing that diffusion features are especially useful for this task and worth using over other faster to obtain features, which I think is lacking. While the paper does this to some degree, and the added supervised results help, as mentioned by the other reviewers, there are many more methods including more supervised (e.g. IFCAT), other generative models, aggregation networks on self-supervised networks, existing methods that extract feature descriptors from diffusion models, etc. that it would be really informative to compare/ablate against to show that the features learned by the diffusion model really are especially useful.
> >
> > To summarise, since the problem of extracting features from diffusion models is of interest and as added in the rebuttal the approach appears to perform well compared to recent supervised approaches, I have increased my score. But this paper would much stronger if we knew better where the improvements come from through a stronger set of baselines/ablations, so I have only increased my score to borderline accept. Happy to discuss with the authors if they disagree with any of these comments.

---

> > > ### Author Response · Authors · 2023-08-16
> > >
> > > We would like to thank the reviewer for taking the time to review our additional results and for providing feedback regarding our evaluation. In Table 9, we ablate the effect of single layer selection, naive concatenation, and training an aggregation network for DINO and DINOv2, symmetric to the ablations we performed for our method. Note that to ensure a consistent evaluation across all backbones, we use the same input resolution of 224. Training an aggregation network noticeably improves the performance of both DINO and DINOv2 by 3\% and 8\% respectively. Interestingly, single layer selection of both DINOv2 and Stable Diffusion features (SD - Layer 4) performs comparably, but training an aggregation network on top of the Stable Diffusion features (i.e., over both timesteps and layers) yields a larger relative boost (14\% PCK\@0.1_img) than DINOv2. **Ultimately, our method that trains an aggregation network on top of Stable Diffusion features performs the best at 72.56\% PCK\@0.1_img, compared with 54.69\% and 68.37\% PCK\@0.1_img for DINO and DINOv2 respectively.** In Table 10 we also verify that our aggregation network on top of DINO features learns mixing weights consistent with the hand-selected features explored in Amir et. al. [3], where it indeed learns that Layers 9 - 11 are particularly useful for the semantic correspondence task. Since IFCAT [4] does not have publicly available code, we display the result reported in the original paper. As such, with these additional baselines we validate that the strong performance is not only from the strong backbone but also the fact that we are aggregating across the layers and timesteps of a diffusion model in particular.
> > >
> > > **Table 9. SPair-71K**
> > > | Model                | PCK\@0.1_img | PCK\@0.1_bbox |
> > > |:---------------------|:-----------:|:------------:|
> > > | IFCAT [4] | - | 64.40* |
> > > | | |
> > > | DINO [3]          |   51.68  | 41.04  |
> > > | DINO - Concat All | 20.17 | 13.60 |
> > > | DINO + Aggregation Network |  54.69  |   44.29  |
> > > | | |
> > > | DINOv2*          |  60.14   |  46.94 |
> > > | DINOv2 - Concat All | 60.89 | 47.69 |
> > > | DINOv2 + Aggregation Network | 68.37 | 56.35|
> > > | | |
> > > | SD - Layer 4 | 58.80 | 46.58 |
> > > | SD - Concat All | 52.12 | 41.83 |
> > > | Ours | **72.56** | **64.61** |
> > > |*Since IFCAT does not have publicly available code, we take the result reported in the original paper.|
> > > |*In our previous reported experiments with DINOv2 in Tables 1-4, we used input images of resolution 770 to account for its large patch size. To ensure a fair evaluation in this table, we use input images of resolution 224 for all methods.|
> > >
> > > **Table 10. DINO + Aggregation Network Learned Mixing Weights**
> > > | Layer | 0 | 1 | 2 | 3 | 4 | 5 | 6 | 7 | 8 | 9 | 10 | 11|
> > > |:---|:---|:---|:---|:---|:---|:---|:---|:---|:---|:---|:---|:---|
> > > | | 4% | 3% | 3% | 3% | 4% | 8% | 5% | 9% | 11% | **14%** | **14%** | **14%**|
> > >
> > > [4] Hong et. al. “Integrative Feature and Cost Aggregation with Transformers for Dense Correspondence.” arXiv 2022.

---

### Official Review · Reviewer_LUzf · 2023-07-10

**Soundness:** 3 good
**Presentation:** 3 good
**Contribution:** 2 fair
**Rating:** 5
**Confidence:** 4

**Summary:**

This paper explores semantic correspondence tasks with stable diffusion model. Specifically, the authors proposed to first extract feature maps varying across timesteps and layers from the diffusion process and trains a lightweight neural network to aggregate them together for semantic correspondence.
Experimental results on CUB-200 and SPair-71k datasets show that proposed method outperforms other baselines.



**Strengths:**


- This paper shows that one could explore diffusion models for semantic correspondence tasks.

- the writing is clear and easy to follow.

- The experiments show that the proposed method could outperform other baselines on the SPair-71k real image benchmark.

- the authors have shown numerous visual examples to demonstrate the correspondence capability.



**Weaknesses:**

1. The proposed method, that "distill the information  distributed across time and space from a diffusion process into a single descriptor map", seems to have more potential than correspondence tasks. Have the authors explore other tasks, like video label propagation, Homography estimation or perception tasks like classification?


2. Concurrent works: there are multiple papers presenting correspondence ability of diffusion models:

[1*] Hedlin, Eric, et al. "Unsupervised Semantic Correspondence Using Stable Diffusion." arXiv preprint arXiv:2305.15581 (2023).
[2*] Tang, Luming, et al. "Emergent Correspondence from Image Diffusion." arXiv preprint arXiv:2306.03881 (2023).
[3*] Zhang, Junyi, et al. "A Tale of Two Features: Stable Diffusion Complements DINO for Zero-Shot Semantic Correspondence." arXiv preprint arXiv:2305.15347 (2023).

Could the authors explain the differences if possible?

**Questions:**

please refer to the weaknesses section.

---

> ### Author Rebuttal · Authors · 2023-08-07
>
> 1. *Have the authors explored other tasks [...]?*
>
> Please see the global response PDF for new applications in semantic appearance transfer and video mask propagation.
>
> 2. *Concurrent works: there are multiple papers presenting correspondence ability of diffusion models: [...] Could the authors explain the differences if possible?*
>
> Indeed there are a few concurrent works that have appeared in unofficial preprints after the submission of this project. **One main conceptual difference with these concurrent works is that we aggregate features across all timesteps of the diffusion process**, which we motivate in Figure 2 of the main paper and ablate in our experimental results (Ours vs Ours - One-Step in Table 1 of the main paper, where we achieve a 10\% boost in PCK\@0.1_bbox). In contrast, these works all use a hand-selected *single timestep.* In Figure 5 of the main paper we demonstrate why heuristics for hand-selecting specific diffusion features may not be generalizable, since SDv1-5 and SDv2-1 display significantly different behavior in which combinations of layers and timesteps are the most “semantic,” as automatically learned by our aggregation network. **Finally, we also report the best keypoint matching performance on SPair-71k (64.6 PCK\@0.1_bbox) compared with 45.4 [1], 52.9 [2], 62.9 [3] PCK\@0.1_bbox respectively.**
>
> [1] Hedlin et. al. "Unsupervised Semantic Correspondence Using Stable Diffusion." arXiv 2023. \
> [2] Tang et. al. "Emergent Correspondence from Image Diffusion." arXiv 2023. \
> [3] Zhang et. al. "A Tale of Two Features: Stable Diffusion Complements DINO for Zero-Shot Semantic Correspondence." arXiv 2023.

---

### Author Rebuttal · Authors · 2023-08-07

### Summary
We thank the reviewers for their helpful feedback and suggestions, which we will integrate into the final manuscript. In this work we present a “simple and [...] very effective” (Reviewer sN43) framework for consolidating the internal representations of a diffusion model for tasks such as semantic correspondence.

To re-state our motivation: prior work shows that features from a hand-selected layer & timestep can be useful for downstream applications. Our work demonstrates that the most useful features are not contained within a single layer and timestep, but rather are distributed across _all layers and timesteps_ of the diffusion sampling process (different choices of time and layer often contain complementary information, as seen in Figure 2 of the main paper). Unfortunately, naive concatenation across all features results in excessively high dimensional descriptors that equally weigh all source features, making distance metrics less practical and useful. Our final proposed approach solves this by aggregating these high-dimensional features into a more useful low-dimensional descriptor map that is trainable for a given target task. **We have performed additional evaluations on PF-WILLOW and PF-PASCAL to demonstrate that our method is transferable. We provide them below, along with new applications in semantic appearance transfer and video mask propagation (see PDF).**

The reviewers agree that “the premise of extracting useful features from diffusion models is an important one” (Reviewer HmuP) and that it is “very relevant” (Reviewer sN43). **Moreover, we clearly demonstrate that we achieve “SOTA performance [...] with large gap to existing works” (Reviewer HmuP), which we further validate with new comparisons to CATS++ and DINOv2 in Table 1 below.**

**Table 1. SPair-71K**
| Model                | PCK\@0.1_img | PCK\@0.1_bbox |
|:---------------------|:-----------:|:------------:|
| DINOv2 [1]            |    68.33    |     56.98    |
| CATS++ [2]           |    70.26    |     57.06    |
| Ours - Indiv. Bottleneck per Timestep       |    73.07    |     65.09    |
| Ours                | **72.56**   | **64.61**   |

**Table 2. CUB**
| Model                | PCK\@0.1_img | PCK\@0.1_bbox |
|:---------------------|:-----------:|:------------:|
| DINOv2 [1]            |    89.96*    |     76.83*   |
| CATS++ [2]           |    75.92    |     59.49    |
| Ours                | **82.29**   | **69.42**  |
|*Please note that DINOv2’s training set included CUB|

### Comparison to CATS++ and DINOv2
**In Table 1, we display results for both CATS++ and DINOv2 on SPair-71k. While these methods are competitive, our method still achieves the best result at 72.56 in PCK\@0.1_img, with a 2\% increase over CATS++ and 4\% increase over DINOv2.** For CUB, PF-PASCAL, and PF-WILLOW (Table 2, 3, 4) we outperform CATS++ by 6\%, 19\%, and 11\% in PCK\@0.1_img respectively. Similarly, DINOv2 performs worse than our method across the board, except for CUB where it exhibits unusually high performance *because it was trained on samples from CUB* (Table 15 of the Appendix in [1]). In Table 5 we display the hyperparameter sweep we conducted for DINOv2 ViT-S/14, leading us to use the Layer 11 token features for our DINOv2 baseline.

### Individual Bottleneck Layer per Timestep Ablation
Our choice to share bottleneck layers is an effort to reduce model size at the cost of a slight performance decrease. **As seen in Table 1, both the method with individual bottleneck layers per timestep and our final method perform similarly, with less than a 1\% degradation in PCK\@0.1_img.** Training a bottleneck layer for each timestep would require 132 projection layers (812.85 MB) compared to just 12 projection layers (26.12 MB) when the bottleneck layers are shared.

**Table 3. PF-PASCAL [3]**
| Model         | PCK\@0.1_img | PCK\@0.1_bbox |
|:---------------------|:-----------:|:------------:|
| DINOv2 [1]    |       84.30 |       78.99 |
| CATS++ [2]    |       68.02 |       62.96 |
| Ours      |    **86.67** |     **82.85** |

**Table 4. PF-WILLOW [4]**
| Model         | PCK\@0.1_img | PCK\@0.1_bbox |
|:---------------------|:-----------:|:------------:|
| DINOv2 [1]    |       86.64 |       71.34 |
| CATS++ [2]    |       78.87 |       66.09 |
| Ours          |       **89.61** |       **77.98** |

### Additional Evaluation Datasets
We focused on SPair-71k and CUB because they presented more complex and varied examples than the other benchmarks, which are largely composed of simple image pairs with “similar viewpoints and scales” [5]. That being said, we are happy to include results on PF-PASCAL and PF-WILLOW (Table 2, 3 above). Note that across all these datasets we transfer the model purely trained on SPair-71k for evaluation. **When transferred to PF-PASCAL and PF-WILLOW, our method outperforms DINOv2 by 2\% and 3\% PCK\@0.1_img respectively.**


**Table 5. DINOv2 Hyperparameter Selection (SPair-71k PCK\@0.1_img)**
| Model                | key | query | value | token |
|:---------------------|:-----------:|:------------:|:------------:|:------------:|
| dinov2_vits14 - Layer 9 |  18.74 | 19.10 | 52.47	| 55.08 |
| dinov2_vits14 - Layer 11 |  36.97 | 36.46 | 64.25 | **68.33** |

**References**\
[1] Oquab et. al. “DINOv2: [...].” arXiv 2023.\
[2] Cho et. al. “CATs++: [...].” TPAMI 2022.\
[3] Ham et. al. “Proposal flow.” CVPR 2016.\
[4] Ham et. al. “Proposal flow: Semantic correspondences from object proposals.” PAMI 2017.\
[5] Min et. al. “SPair-71k: [...].” arXiv 2019.

---

### Decision · Program_Chairs · 2023-09-21

**Decision:**

Accept (poster)

**Comment:**

The authors' extensive rebuttal has addressed the critical points raised by the reviewers. After considering the rebuttal, discussions, and also taking the authors' additional comments into account, all reviewers unanimously agreed to accept the paper -> accept. Congratulations!
Please take the reviewer comments into account when preparing your camera-ready version.